# The Geometric Origin of Grokking:
# Accelerating Generalization via Active Structural Reorganization

Kefei Tao [* 1]   Zhang Zhang [* 2]   Mingze Qi [1]   Xiaojun Duan [1]

## Abstract

Grokking, the phenomenon where models suddenly generalize long after overfitting training data, remains a puzzling challenge in neural network dynamics. Through mechanistic analysis, we find that this transition is fundamentally driven by a structural reorganization of token representations, with the onset of grokking entailing a shift toward a well-defined geometry, and reveal the model's distinct understanding of data's dual characteristics. Building on these geometric insights, we propose R2G (Repel-to-Grokking) Loss, an active intervention that reshapes the representation manifold by enforcing structural repulsion. The versatility of R2G is empirically validated in both algorithmic and linguistic tasks, while our theoretical analysis and ablation studies jointly demonstrate that angular reorganization is the primary driver of grokking. Our work offers a novel mechanistic perspective on the evolution of grokking and provides a useful tool for enhancing model efficiency and reliability.

## 1. Introduction

While artificial neural networks have achieved remarkable success, their internal generalization dynamics remain poorly understood. A particularly intriguing phenomenon is grokking—where models abruptly transition from memorization to perfect generalization long after overfitting the training data (Power et al., 2022). While prior research has replicated this transition in various domains (Chughtai et al., 2023; Liu et al., 2023; Murty et al., 2023), a fundamental question remains: What internal dynamics drive the model

grokking?

Existing literature on grokking has largely bifurcated into two directions. The first focuses on mechanistic interpretability, such as the "clock" or "pizza" algorithms (Nanda et al., 2023; Zhong et al., 2023) and the circuit competition model (Varma et al., 2023), providing post-hoc explanations for specific tasks. The second focuses on temporal acceleration, aiming to minimize the number of training iterations required for generalization through optimization tricks like gradient amplification or weight norm constraints (Liu et al., 2023; Lee et al., 2024; Xu et al., 2025).

However, we argue that the primary bottleneck in grokking research is not just the time it takes to generalize, but the phase evolution dynamics of generalization level itself, which are influenced by hyperparameters like data size and model hidden size. Huang et al. categorized them into four phases: Progression, Memorization, Semi-Grokking and Grokking (Huang et al., 2024). Prior work has noted that generalization correlates with the formation of structured representations (Liu et al., 2022; He et al., 2024), but have not focused on how internal representations drive the transition between these phases, which is crucial for extracting features from data and completing tasks.

In this paper, we conduct a mechanistic analysis of the internal representation space across phases within the attention layers of Transformers. We identify that the phase transition is fundamentally driven by structure reorganization, revealing the model's distinct understanding of data's dual characteristics and proposing a novel structural perspective on understanding grokking. Based on these insights, we propose R2G (Repel-to-Grokking) Loss, which acts as an active structural intervention. By enforcing geometric repulsion in the embedding space, it facilitates the emergence of higher-level generalization. Our empirical results across multiple arithmetic and linguistic benchmarks demonstrate that R2G achieves superior performance under identical data and computational budgets (Figure 1). In addition, we develop a theoretical analysis of representation dynamics and perform controlled ablation studies that isolate the roles of norm and angular components, jointly validating R2G mechanism. Our contributions are summarized as follows:

---
[*]Equal contribution  [1]College of Science, National University of Defense Technology, Changsha Hunan, China [2]Department of Physics, Centre for Nonlinear Studies, Hong Kong Baptist University, Hong Kong SAR, China. Correspondence to: Mingze Qi <qimingze17@nudt.edu.cn>.

*Proceedings of the 43rd International Conference on Machine Learning*, Seoul, South Korea. PMLR 306, 2026. Copyright 2026 by the author(s).

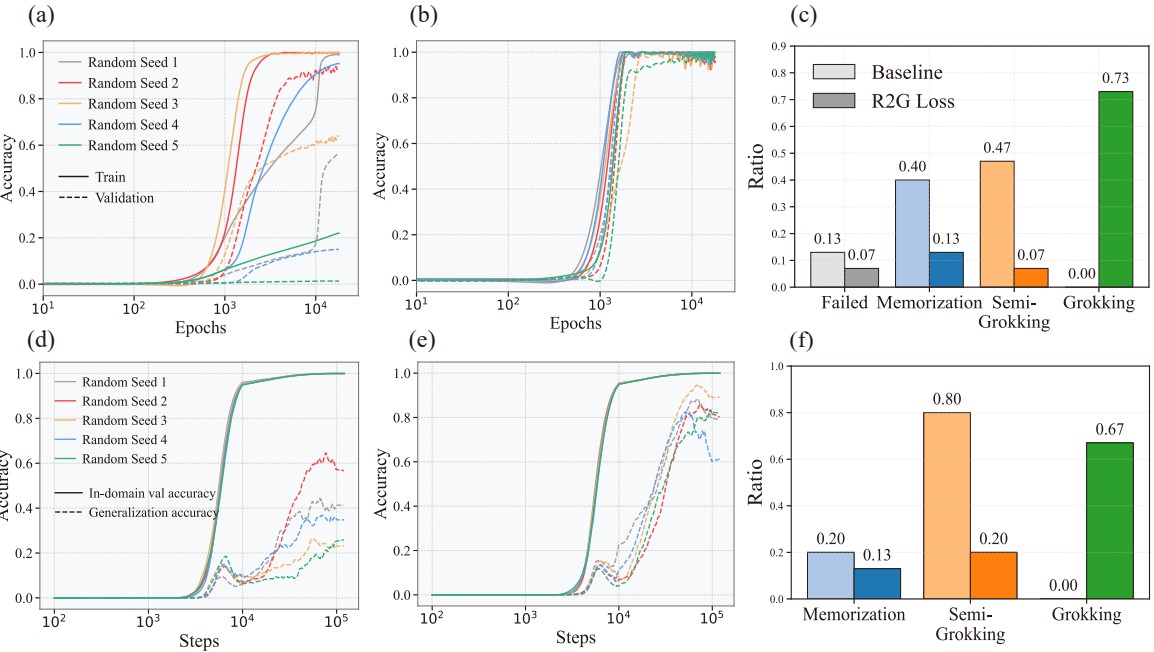

*Figure 1.* The R2G loss accelerates grokking. (a-c) The modular $P = 500$ arithmetic task. (d-f) The tense-inflection task. Under identical input data, the (a,d) original training curves, and (b,e) R2G loss training curves after smoothing show that different original training states are successfully promoted to achieve grokking after applying R2G Loss method. (c,f) The improvement brought by R2G loss over baseline across four distinct tasks: Failed, Memorization, Semi-grokking, and Grokking, the vertical axis Ratio represents the proportion occupied by each phase. Notably, R2G loss demonstrates significant gains in the Grokking category.

- **Geometric Characterization in Phase Evolution**: We track the structural evolution of representations across different training phases, revealing the emergence of a structured geometric manifold during the phase transition; And interpret the grokking mechanism through the dual characteristics (symbolic and numerical) inherent in the data.

- **R2G Loss as an Active Structural Intervention**: We propose the R2G (Repel-to-Grokking) loss. R2G serves as a targeted intervention that drives models out of the non-grokking phase under identical data and optimization conditions.

- **Theoretical and Empirical Validation of R2G Mechanism**: We validate the effectiveness of R2G across diverse arithmetic and linguistic tasks. Theoretical analysis and a series of controlled ablation studies further demonstrate the effectiveness of the R2G design.

## 2. Related Work

### 2.1. Interpretability of Grokking

A core direction in grokking research is reverse-engineering the learned algorithms within the model's parameters. Notable findings include the "clock" or "pizza" algorithms,

which reveal that Transformers solve modular tasks by projecting token embeddings onto a circular topology using discrete Fourier transforms (Nanda et al., 2023; Zhong et al., 2023). Furthermore, the emergence of generalization is found to be highly correlated with the formation of structured representations (Power et al., 2022; Liu et al., 2022). Gromov (Gromov, 2023) and He (He et al., 2024) provide complementary geometric analyses of grokking representations in modular arithmetic and in-context learning settings, respectively. However, these interpretations are largely post-hoc, explaining what the model has learned rather than how to actively guide the learning process toward these structured states. While Varma et al. attribute grokking to the competition between memorization and generalization circuits (Varma et al., 2023), we argue that this competition can be decisively biased toward generalization by actively reshaping the representation manifold. We bridge this gap by introducing an active intervention that facilitates the necessary structure reorganization, extending prior geometric observations to a causal and interventional framework.

### 2.2. Acceleration of Grokking

Existing methods to eliminate grokking primarily focus on temporal acceleration. For instance, Grokfast employs a low-pass filter to amplify slow, low-frequency gradients that

drive generalization (Lee et al., 2024). Other approaches include constraining weight norms within a spherical radius (Liu et al., 2023), initializing with pre-grokked weights (Furuta et al., 2024), amplifying specific gradient components (Lee et al., 2024), applying lottery ticket masks (Minegishi et al., 2025), and GrokTransfer achieves delay reduction without pre-trained models or additional data (Xu et al., 2025). While effective at reducing latency, these methods are essentially temporal heuristics only focus on grokking phase that do not address the fundamental structural cause of non-grokking runs. Instead of merely speeding up the clock, our work actively reshapes the representation geometry to accelerate phase transition, transforming stagnant memorization into successful generalization.

### 2.3. Loss Function Design

Beyond standard objective functions like Cross-Entropy, designing auxiliary losses to shape the geometry of the representation space has proven to be effective for OOD detection (Ming et al., 2022) and sampling diversity (Wang & He, 2025). However, most existing regularization techniques, such as weight decay, exert a passive constraint on parameter norms without explicitly considering the topological structure of the embedding manifold. Our proposed R2G loss distinguishes itself by acting as an active structural intervention to facilitate the phase transition into grokking.

## 3. Problem and Experimental Setup

### 3.1. Task

Research on grokking originated from and has been predominantly based on modular arithmetic tasks (Power et al., 2022; Nanda et al., 2023), as they offer a controlled and computationally efficient setting for observing the phenomenon and conducting mechanistic analysis. Accordingly, we adopt modular addition as our primary benchmark:

$$(a + b) \bmod P = c, \tag{1}$$

where $a, b \in (0, \ldots, P - 1)$, $P \in \mathbb{N}$, a model processes inputs $\{a, +, b, \%, P, =\}$, and $a, b \in (0, 1, 2, \ldots, P - 1)$, to predict $c \equiv a + b (\bmod P)$. Our experiments encompass cases where $P$ is both prime and composite. $P = 500$ was chosen for presentation in the main text, because a larger modulus provides a broader scope for investigation, where the exhibited patterns demonstrate greater generality. In later sections, we also present experimental results for other moduli $P$ and different operations.

These tasks are viewed as classification tasks, where the label number is $P$. We split training and validation subsets by employing distinct $(a, b)$ tuple combinations. For simplicity, we directly use validation performance to distinguish memorization, semi-grokking, and grokking. More

comprehensive data details are available in Appendix A.2. In the grokking acceleration part, we also validate the R2G loss method on a language processing task in Section 5.3.

### 3.2. Four Phases of Generalization

The model's performance can be divided into different phases. And under a fixed hidden size, progressively increasing train data size improves the generalization level of models, leading to a phase evolution from memorization to semi-grokking to grokking, which is the primary object of our study. In this paper, the four phases are defined according to the models' train-test performance gaps:

- **Failed**: Even after a sufficiently long training period, the training accuracy still does not reach a perfect level.

- **Memorization**: Memorization dominates, with high training accuracy and low validation accuracy.

- **Semi-Grokking**: Comparable efficiency allows partial generalization post-memorization.

- **Grokking**: Generalization dominates, yielding near-perfect validation accuracy.

### 3.3. Model

For the modular arithmetic analysis in the main text we employ a one-layer decoder-only Transformer (Vaswani et al., 2017) with 4 attention heads and a hidden dimension of $d_{\text{model}} = 48$, utilizing Gaussian Error Linear Unit (GeLU) activations. The model is trained using the AdamW optimizer (Loshchilov & Hutter, 2019) with a learning rate of $10^{-3}$ and a weight decay coefficient of $\gamma = 10^{-2}$. For the optimization objective, we adopt the standard cross-entropy loss for next-token prediction. And the tense-inflection experiment in Section 5.3 uses a different architecture, detailed settings are presented in Appendix B.1

To systematically investigate the phase transitions from memorization to grokking, we vary the training dataset size from 8,000 to 40,000 samples, while maintaining a fixed validation set of 2,000 samples. This setup ensures that our experiments cover the full spectrum of generalization dynamics, including the transition through the semi-grokking phase. Further details on data generation are provided in Appendix A.1.

## 4. Grokking Mechanistic Analysis

### 4.1. Symbolic Representation Analysis: QKV Pattern

In Transformers, the Query ($Q$), Key ($K$), and Value ($V$) vectors constitute the fundamental elements of the attention mechanism, each fulfilling a distinct functional role. To in-

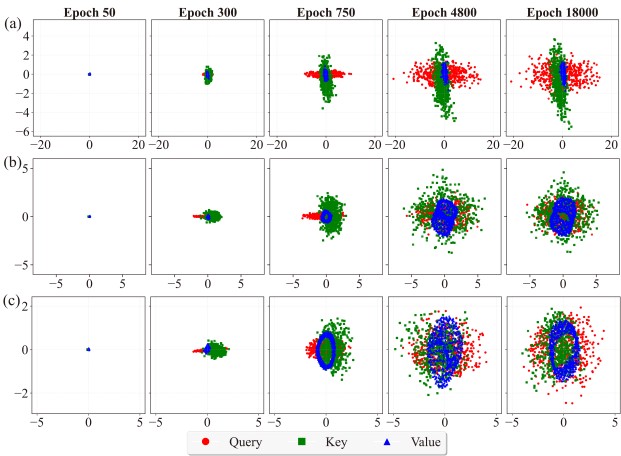

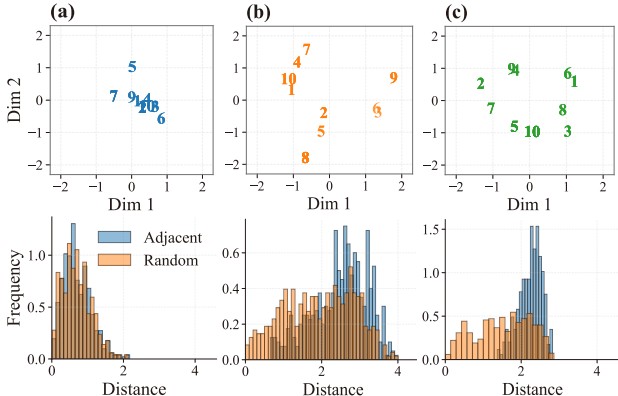

*Figure 2.* Representation of $Q, K, V$ vectors of 2000 validation data in a two-dimensional space after PCA, and five representative epochs are selected for each case. (a) memorization ($N = 8000$). (b) semi-grokking ($N = 28000$). (c) grokking ($N = 40000$). The figure exclusively illustrates the results corresponding to operand $a$, while results for $b$ and $c$ are detailed in Appendix C.1.

vestigate the fine-grained dynamics of grokking, we analyze the evolution of these vectors independently:

**Query** ($Q$): Query vectors ($Q = XW_Q$) encode the information demand of a token, representing "what the token seeks to know" from the context.

**Key** ($K$): Key vectors ($K = XW_K$) act as addressable descriptors answering "what this token can provide". The attention weights are determined by the alignment scores:

$$\text{Attn}(Q, K) = \text{softmax}\left(\frac{QK^T}{\sqrt{d_{\text{model}}}}\right). \tag{2}$$

**Value** ($V$): While $Q$ and $K$ govern attention allocation, the value vectors ($V = XW_V$) define "what information to transmit". Visualizing $V$ vectors provides a direct window into the model's internal representation manifold.

To reveal latent structural patterns in high-dimensional representations, we apply Principal Component Analysis (PCA), which preserves geometrically meaningful variance, unlike nonlinear methods such as t-SNE that may distort global distances. As shown in Figure 2, representations exhibit distinct geometric behaviors across training phases: during memorization, vectors collapse into compact clusters, whereas during grokking they become well separated. Notably, value vectors organize into circular structures in the projected space, corresponding to hyperspherical arrangements in higher dimensions. This observation motivates our core hypothesis: *the emergence of grokking correlates with increased pairwise distances between representation vectors*.

*Figure 3.* Row1: The embedding position of token 1~10 of operand $a$ (same for the results of operand $b$ and $c$ in Appendix C.3); Row2: Histogram of Euclidean distances between 48-dimensional representations on validation data (blue) and random pairs (red). (a) memorization (b) semi-grokking (c) grokking.

### 4.2. Numerical Representations Analysis: Adjacency Repulsion

To investigate our hypothesis, we visualize the value vectors ($V$) of operands. While we focus on tokens $1 \sim 10$ of operand $a$, tokens in other ranges (e.g., $11 \sim 20, 21 \sim 30$) and other operand $b$ and $c$ exhibit consistent geometric patterns, as detailed in Appendix C.2.

As illustrated in Figure 3, the representation geometry undergoes a profound transformation. During memorization and semi-grokking, representations are distributed in a stochastic manner; whereas in the grokking phase, they form a circular topology where numerically adjacent digits are positioned with maximal separation. The distance histograms in Figure 3 quantitatively validate this observation:

- **Memorization**: Adjacent-digit distances (red) match random digit pairs (blue), confirming random representations;

- **Semi-Grokking**: The adjacent-number distribution begins to diverge from the random baseline, signaling the onset of structural emergence;

- **Grokking**: Adjacent numbers exhibit systematically larger distances than random pairs, reflecting a structured repulsion in the latent space.

From the perspective of a model devoid of prior knowledge, input data exhibits dual characteristics: numerical magnitude and symbolic identity. The model forms a specific structure in the representation space by understanding the relationships among numerical values. Our findings reveal that grokking is characterized by the model's transition from treating tokens as isolated symbols to comprehending their numerical relationships.

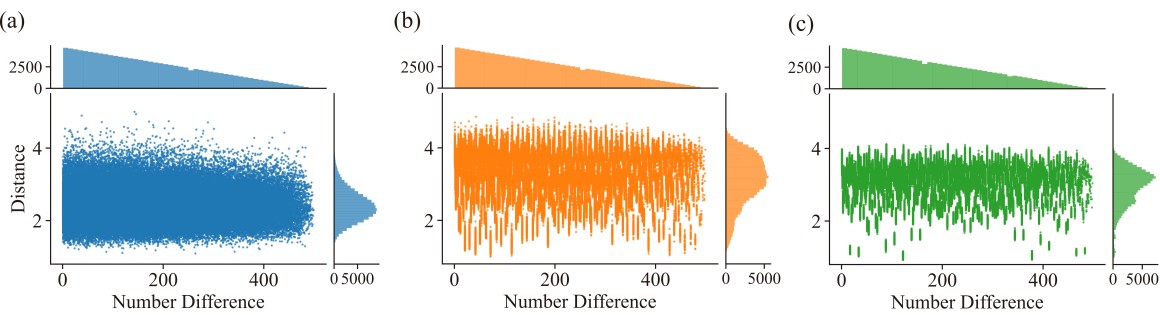

*Figure 4.* Scatter plot of the original numerical difference (1~499) and the 48-dimensional spatial distance. (a) Memorization($N = 8000$), (b) Semi-grokking ($N = 28000$). (c) Grokking ($N = 40000$).

To quantify the extent of this numerical understanding, we analyze the correlation between the raw numerical difference of tokens and their 48D embedding distance (Figure 4). Key observations include:

- **Memorization**: Embedding distances remain random regardless of numerical differences;

- **Semi-Grokking**: Partial discrimination of numerical differences but more dispersed distance distributions;

- **Grokking**: The model achieves a concentrated distance distribution, successfully mapping the relationship of numerical invariants into geometric distances.

The transition from a dispersed to a concentrated distribution signifies the model's shift from stochastic lookup to deterministic rule-following, evidence that the model has largely internalized the intrinsic numerical features.

## 5. R2G Loss Accelerates Grokking

Building upon our mechanistic findings in Section 4, we propose the Repel-to-Grokking (R2G) Loss framework. Our objective is to determine whether active geometric intervention can catalyze this phase transition.

### 5.1. R2G Loss Formulation

Loss functions based on task-specific numerical priors would limit the method's applicability. To ensure generality, we focus exclusively on symbolic geometric priors. We inject the circular structure of value vectors in the grokking phase into the optimization process by actively increasing the spatial separation between representation vectors (Figure 5). Specifically, we define the R2G loss by maximizing the Euclidean distance between randomly sampled pairs of Value ($V$) vectors within the attention mechanism.

Formally, let $\mathcal{V} = \{v_1, v_2, \ldots, v_n\}$ be the set of $V$ vectors

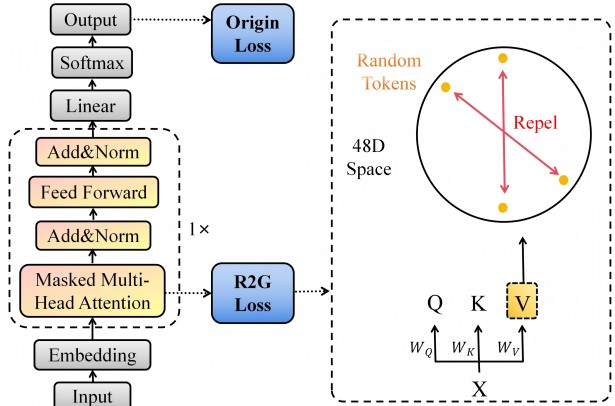

*Figure 5.* Diagram of R2G Loss: it accelerates grokking by enforcing structural separation of random $V$ vectors in 48D hidden space.

in a latent space for a given batch. We define the R2G loss as:

$$\mathcal{L}_{R2G} = \frac{1}{\sum_{i=1}^{n} \|v_i - v_i'\|_2} \tag{3}$$

where $v_i'$ represents a vector from a randomly shuffled version of $\mathcal{V}$, ensuring that the loss acts on arbitrary pairs of vectors. The total optimization objective is formulated as:

$$\mathcal{L} = \mathcal{L}_{\text{orign}} + \alpha \cdot \mathcal{L}_{R2G}, \tag{4}$$

where $\mathcal{L}_{\text{orign}}$ is the standard cross-entropy loss, and $\alpha$ is a tunable hyperparameter that governs the strength of the repulsive force. When $\alpha = 0$, the framework reverts to the baseline model. The selection criteria and robustness for $\alpha$ are detailed in Appendix B.2.

The R2G framework is remarkably lightweight: it requires no additional data, introduces no new learnable parameters,

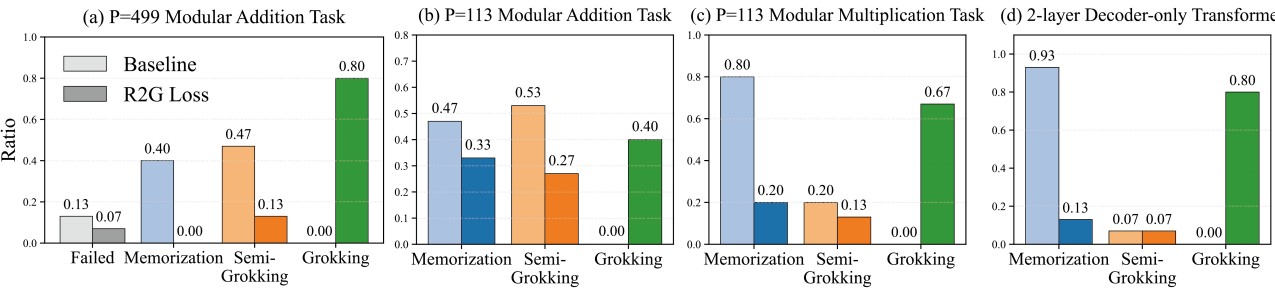

*Figure 6.* The changes brought by R2G loss over the baseline of three phases. (a) The result for the $P = 499$ modular addition task. (b)The result for the $P = 113$ modular addition task. (c) The result for the $P = 113$ modular multiplication task. (d) The result for the $P = 113$ modular addition task on 2-layer decoder-only transformer.

---

**Algorithm 1** R2G Loss

---

**Input:** $V_a, V_b, V_c \in \mathbb{R}^{B \times D}$ ($B$ = batch size, $D$ = hidden dim, Scaling factor $\lambda$)
**Output:** $L_{\text{R2G}} \in \mathbb{R}$
$perm \leftarrow \text{RandomPermutation}(B)$
**for** $i \in \{a, b, c\}$ **do**
  $V_i' \leftarrow V_i[perm, :]$
**end for**
$L_{\text{R2G}} \leftarrow \dfrac{\lambda}{\sum_{i \in \{a,b,c\}} \|V_i - V_i'\|_2}$
**return** $L_{\text{R2G}}$

---

and relies solely on intermediate representations. As outlined in Algorithm 1, this plug-and-play simplicity allows R2G to be seamlessly integrated into Transformer architectures. In later sections, we validated the effectiveness of the R2G framework across both symbolic mathematics and natural language processing tasks.

### 5.2. Performance on Arithmetic Tasks

For modular arithmetic tasks, R2G loss substantially accelerates the grokking process. The main experiment on modular addition with $P = 500$ is shown in Figure 1(a,b,c). Under a fixed training dataset of 26,650 samples, R2G loss significantly elevates the proportion of non-grokking runs reaching full grokking by 73%. Specifically, this improvement is attributed to a successful phase transition of 40% from the semi-grokking stage, 27% from the memorization phase, and 6% rescued from previously failed regimes, respectively. Notably, we observe that R2G loss is capable of not only advancing semi-grokking models to grokking but also rescuing runs that fail to train, effectively elevating a portion of Failed phase into the Grokking phase.

To verify the consistency of our method, we extended the evaluation to prime modular addition tasks ($P = 499$ and $P = 113$). In both settings, R2Gloss consistently evinced

superior acceleration. As shown in Figure 6 (a), for $P = 499$ with 26,600 data, R2G loss increases the grokking by 80%, while reducing the failed by 6%, memorization phase by 40% and the semi-grokking phase by 34%. Similarly, for $P = 113$ with 2,650 data (Figure 6 (b)), the grokking rate improves by 40%, and decreases the memorization and semi-grokking phases by 14% and 26%, respectively.

We further evaluate R2G loss on $P = 113$ modular multiplication tasks ($a \times b\%p = c$) to assess its generality. As reported in Figure 6 (c), with a dataset size of 2,500, the grokking rate increases by 67%. The results collectively demonstrate that R2G loss serves as a universal catalyst for grokking, effectively driving models across the generalization gap regardless of the specific modular operation.

R2G also generalizes across architectures: on a 2-layer decoder-only Transformer ($P = 113$, 3,700 data), it lifts the grokking rate by 80% (Figure 6(d)). Furthermore, sweeping $\gamma \in \{0, 0.001, 0.01, 0.1\}$ confirms that R2G consistently outperforms the baseline regardless of weight decay, indicating an independent geometric benefit (Appendix G).

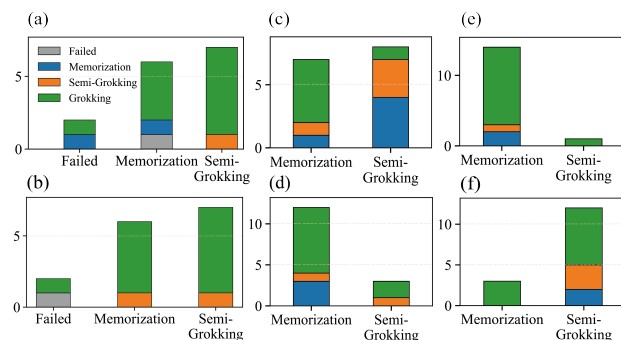

*Figure 7.* Phase transitions facilitated by R2G loss. X-axis: original phases, y-axis: the times of transition to each phase. (a,b,c) $P = 500, 499, 113$ addition task. (d) $P = 113$ multiplication task. (e) Experiment on 2-layer decoder-only transformer. (f) Tense-inflection task.

## 5.3. Cross-Domain Validation: Linguistic Task

To evaluate the broader applicability and generalizability of R2G, we extended our experiments to a Tense-Inflection task, a language processing task that leverages English subject–verb agreement to reveal a model's syntactic generalization ability (Linzen et al., 2016). Since the statements have a hierarchical structure, structural grokking will occur in this task (Murty et al., 2023). Here we determine whether structural grokking occurs based on the model's performance on the out-of-distribution test set, where the model continues to acquire hierarchical structures long after the in-domain validation performance has saturated. The detailed description of this task is provided in Appendix D.2.

As illustrated by the training curves in Figure 1(d,e), the introduction of R2G loss drastically alters the learning dynamics. With a fixed dataset with 10,000 samples, R2G elevates the grokking success rate by 67%, the proportion of semi-grokking and memorization decreases by 7% and 60%, as shown in Figure 1(f). The specific model configuration for this task is detailed in the Appendix B.1. To further refine our results, we present the proportions of each phase transition of 5 tasks in Figure 7. These results highlight the effectiveness of R2G loss in shifting models out of low generalization, demonstrating its role as a general method for grokking in diverse tasks.

The effectiveness of R2G loss may arise from its ability to promote token representation disentanglement and better structure. Tasks usually require precise symbolic encoding, our loss function actively guides the network to enhance inter-token separability, which in turn amplifies discriminative information for token identity and induces the emergence of intrinsic structures within the model's latent space.

### 5.4. Comparison with Other Acceleration Methods

Prior work on accelerating grokking, such as Grokfast (Lee et al., 2024) and NeuralGrok (Xu et al., 2025), primarily operates along the *temporal* axis. In contrast, R2G is motivated by a structural, geometric perspective: it actively reshapes the representation manifold to catalyze the phase transition. Despite this difference in motivation and mechanism, we conduct empirical comparison to assess R2G's practical advantages under a unified evaluation protocol.

We compared with Grokfast and NeuralGrok methods on the same modular addition task ($P = 500$) under identical training data (dataset size = 26,650). Grokfast applies an exponential moving average filter to amplify slow, low-frequency gradient components ($\alpha = 0.3$, task-optimal); NeuralGrok trains an auxiliary amplifier that adaptively re-weights gradients ($c_{\text{norm}} = 0.2$, *innerloopsteps* $= 2$, task-optimal).

As shown in Figure 8, all three methods improve upon the baseline, yet R2G achieves the highest grokking success rate of **73%**, compared to 60% for Grokfast and 27% for NeuralGrok. Crucially, Figure 8(b) reveals that R2G also converges the fastest among successfully grokking runs, demonstrating a consistent advantage along both the phase axis and the temporal axis. These results suggest that geometric restructuring of the value representation manifold addresses a more fundamental bottleneck.

## 6. Theoretical analysis

### 6.1. Radial–Angular Decomposition

The R2G loss increases the pairwise distance between representation vectors $\|v_i - v_j\|$, which can be achieved either by expanding angular separation or by scaling up vector magnitudes in high-dimensional manifolds. Motivated by the success of R2G, we analyze the gradient structure by separating the radial and angular components of representations.

We consider a standard multi-class classification setting. Let $v \in \mathbb{R}^d$ denote the representation of an input, and let $W = \{w_1, \ldots, w_K\}, w_k \in \mathbb{R}^d$ be the classifier weights. The logits are given by $z_k = w_k^\top v$, and the predicted probabilities follow the softmax distribution

$$p_k = \frac{e^{z_k}}{\sum_{j=1}^{K} e^{z_j}}. \tag{5}$$

Given the ground-truth label $y$, the cross-entropy loss is $\mathcal{L}(v) = -\log p_y$. The gradient of the loss with respect to the logits satisfies

$$\frac{\partial \mathcal{L}}{\partial z_k} = p_k - \mathbf{1}[k = y]. \tag{6}$$

By the chain rule, the gradient with respect to the representation $v$ is

$$\nabla_v \mathcal{L} = \sum_{k=1}^{K}(p_k - \mathbf{1}[k = y])w_k = \sum_{k=1}^{K} p_k w_k - w_y. \tag{7}$$

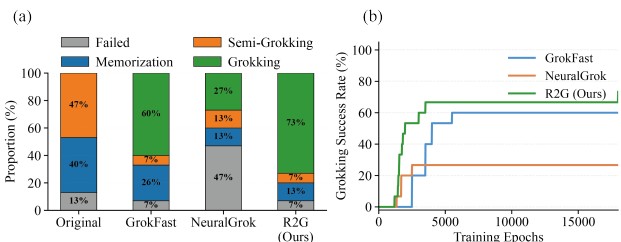

*Figure 8.* Comparison of grokking acceleration effects between R2G loss, Grokfast, and NeuralGrok. (a) Proportions of failed / memorization / semi-grokking / grokking states after applying different methods and the baseline, on P=500 modular addition task under datasize 26650. (b) Grokking rate at different epochs for different methods.

We denote this effective driving force as

$$g \triangleq \sum_{k=1}^{K} p_k w_k - w_y, \tag{8}$$

so that $\nabla_v \mathcal{L} = g$. Importantly, $g$ aggregates contributions from all classes, implying that representations updates are inherently collective rather than class-local.

We decompose the representation into radial and angular components:

$$v = ru, \quad r = \|v\|, \quad \|u\| = 1. \tag{9}$$

Under this parameterization, the logits become $z_k = r\,(w_k^\top u)$, showing that the softmax input depends on both the representation norm and its directional alignment with the classifier weights. We consider gradient descent updates

$$v_{t+1} = v_t - \eta \nabla_v \mathcal{L}. \tag{10}$$

Let $\Delta v = -\eta g$ denote the update increment. The induced updates on the radial and angular variables admit an exact decomposition: $\Delta r = u^\top \Delta v$, $\Delta u = \frac{1}{r}(I - uu^\top)\Delta v$. Substituting $\Delta v = -\eta g$, we obtain

$$\Delta r = -\eta\, u^\top g, \tag{11}$$

$$\Delta u = -\frac{\eta}{r}(I - uu^\top)g. \tag{12}$$

The above equations reveal a fundamental asymmetry in learning dynamics. The radial update $\Delta r$ is not geometrically suppressed and can evolve rapidly, whereas the angular update $\Delta u$ is scaled by $1/r$, causing directional learning to slow as the representation norm grows. Consequently, standard cross-entropy training induces a natural separation of time scales: the representation norm $r$ acts as a fast variable that quickly reduces training loss, while the representation direction $u$ evolves slowly and governs representation geometry. This separation emerges directly from the gradient structure of the softmax cross-entropy objective.

This distinction has important implications for grokking. We therefore interpret grokking as being driven by the slow, collective reorganization of angular variables: models may remain in prolonged memorization phases through radial scaling alone, and generalization emerges only once the angular geometry reorganizes into a coherent structure.

This perspective yields clear, testable predictions: interventions that primarily affect representation norms should have limited impact on grokking, whereas interventions targeting angular structure should be substantially more effective. This inference is also validated by our ablation studies in Section 7.1.

## 6.2. Reciprocal Form of R2G Loss

The R2G loss in Equation (3) minimizes the reciprocal of the total pairwise separation, thereby maximizing the aggregate distance between Value representations. This reciprocal form enables stable and effective geometric intervention.

**Adaptive Gradient Scaling.** Let $D = \sum_i \|v_i - v_i'\|_2$ denote the total distance. The gradient of $\mathcal{L}_{R2G} = \lambda/D$ with respect to an individual representation $v_k$ satisfies:

$$\nabla_{v_k} \mathcal{L}_{R2G} = -\frac{\lambda}{D^2} \cdot \frac{\partial D}{\partial v_k}. \tag{13}$$

The key property lies in the $1/D^2$ prefactor: when representations are collapsed (small $D$, characteristic of the memorization phase), this factor is large, automatically amplifying the repulsive force precisely when structural reorganization is most needed. Conversely, as representations become well-separated (large $D$, approaching the grokking phase), the gradient magnitude diminishes, preventing disruption of the already-formed geometric structure. This self-regulating behavior is a direct consequence of the reciprocal form.

In contrast, consider the quadratic form $\mathcal{L}_{sq} = -\lambda \sum_i \|v_i - v_i'\|_2^2$. Its gradient satisfies:

$$\nabla_{v_k}(-\lambda D^2) \propto -\lambda(v_k - v_k'), \tag{14}$$

whose magnitude grows proportionally with distance. This is counter-productive: in the grokking phase, when representations are already well-dispersed, the repulsive force continues to grow, risking the destruction of the ordered circular topology that has emerged.

We also provide a detailed theoretical analysis of the random permutation mechanism of R2G Loss in Appendix E, and causal evidence of our geometric hypothesis in Appendix F.

## 7. Ablation Study

### 7.1. Interventions on Radial and Angular Dynamics

To validate the theoretical predictions derived in the previous section, we perform controlled interventions on the radial and angular components of representation updates. Specifically, we compare the grokking ratio and convergence speed of the R2G loss with its decoupled variants: Norm-only and Angular-only, on datasets that fail to grok under the baseline setup.

**Norm-only Loss**: Targets only the $L_2$ norm, forcing vectors away from the origin without constraining their relative orientations:

$$L_N = \frac{1}{\sum_{i=1}^{n} \|v_i\|_2} \tag{15}$$

*Table 1.* Ablation study of R2G loss components

| Method | Mean Epoch (90%) | Improve Ratio | Grokking Ratio |
|---|---|---|---|
| Origin | N/A | N/A | 0 |
| R2G loss | 1882 | 80% | 65% |
| Angular | 2500 | 75% | 55% |
| Norm | 2500 | 70% | 45% |

**Angular-only Loss**: Focuses exclusively on hyperspherical topology by maximizing the Euclidean distance between $L_2$-normalized vectors:

$$L_A = \frac{1}{\sum_{i=1}^{n} \| \frac{v_i}{\|v_i\|} - \frac{v_i'}{\|v_i'\|} \|_2} \tag{16}$$

Algorithm of the two loss functions are detailed in Appendix H.1, these decoupled variants maintain the simplicity same as R2G loss. As summarized in Table 1, compared to non-grokking in the baseline setup, the synergy of direction and norm in R2G yields the most robust performance. R2G successfully rescues 65% of the non-grokking runs, outperforming Angular (55%) and Norm (45%).

The poor rescue rate of $L_N$ confirms that radial scaling alone is insufficient, while the stronger performance of $L_A$ highlights the critical role of angular geometry. R2G outperforms both decoupled variants, also reduces mean epochs to grokking by 24.7%, suggesting that jointly coordinating radial and angular dynamics yields a more efficient generalization trajectory.

### 7.2. Comparison of R2G Loss Formulations

We conduct a controlled comparison across loss formulations to empirically validate the necessity of the reciprocal form. Beyond the decoupled Norm-only and Angular-only variants studied in Section 7.1, we evaluate direct alternatives that replace the reciprocal with two normalization-based additive objectives: **LayerNorm** and **Hyperball**. Full algorithmic details of both variants are provided in Algorithms 5 and 6 in Appendix H.2. As reported in appendix Figure 17, LayerNorm achieves a grokking ratio of 60%, which is 13% below R2G (73%); Hyperball attains only 33%, which is 40% below R2G. These results collectively confirm that the reciprocal form of R2G is not interchangeable with alternatives, and that its adaptive gradient scaling property is the key factor driving its superior performance.

### 7.3. Ablation Studies on Key and Query Vectors

The design of R2G is motivated by the hypothesis that grokking arises from structural reorganization of the Value ($V$) manifold rather than changes in attention scoring alone. To verify that R2G targets the most influential component, we apply the loss to $Q$ and $K$ as a controlled ablation. As

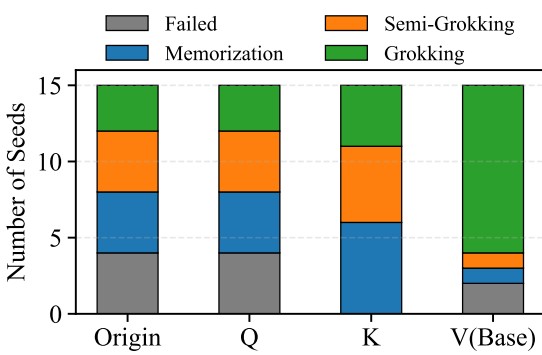

*Figure 9.* Ablation study on the sensitivity of attention components ($Q$, $K$, and $V$ vectors) to the R2G loss.

shown in Figure 9, intervening on $Q$ or $K$ yields success rates near the baseline, whereas intervening on $V$ consistently rescues a substantial fraction of non-grokking runs, confirming that R2G targets a critical bottleneck in representation learning.

## 8. Conclusion

Grokking provides a unique window into the internal dynamics of neural network generalization. We show that varying training data induces transitions from memorization to grokking, driven by qualitative changes in representation geometry. Analysis of Transformer attention embeddings demonstrates that grokking corresponds to an active structural reorganization of the embedding space, shifting from collapsed representations to a decoupled geometric manifold. Examination of value vectors further indicates a transition from symbolic lookup to the internalization of numerical structure.

Based on these insights, we propose R2G (Repel-to-Grokking) Loss, a task-agnostic geometric intervention that promotes structural separation among value vector representations. Through theoretical analysis and controlled ablation studies, we show that angular reorganization—rather than norm growth alone—is the primary driver of grokking. Experiments on arithmetic and linguistic tasks demonstrate that R2G consistently improves generalization under identical data and computational budgets.

Extending R2G to more complex real-world settings and exploring its applicability beyond classical grokking regimes remain promising directions for future work. As a preliminary step, we examined V-vector structural reorganization in OLMo-2-1B across pretraining checkpoints (Appendix I), our findings suggest that geometric reorganization may be a general phenomenon in large-scale pretraining. All code and datasets are available at https://github.com/taokefei/R2G_grokking.

## Impact Statement

This paper presents work whose goal is to advance the field of Machine Learning by exploring the geometric origins of generalization. Our findings on representational reorganization contribute to the fundamental understanding of neural network dynamics. By introducing the R2G loss, our research offers a principled method to accelerate generalization in grokking phenomena. The potential societal consequences include: (1) Cost Efficiency: Reducing the training data required for models to reach better generalization, thereby lowering the carbon footprint of large-scale AI development. (2) Model Reliability: Enhancing the transparency of internal representations, which is a critical step toward building more predictable and trustworthy AI systems. There are no specific ethical concerns or negative societal impacts that we feel must be highlighted here.

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

# A. Dataset

### A.1. Data Used for Grokking/Semi-grokking/Memorization

The main experiments in this paper are conducted on the following three datasets:

- Memorization: training data size $N = 8000$ (Figure 10 (a))

- Semi-grokking: training data size $N = 28000$ (Figure 10 (b))

- Grokking: training data size $N = 40000$ (Figure 10 (c))

For each operation, we construct a dataset of equations with the form $\langle a \rangle \langle op \rangle \langle b \rangle \langle \% \rangle \langle P \rangle \langle = \rangle \langle c \rangle$, where $\langle x \rangle$ stands for the token corresponding to element x.

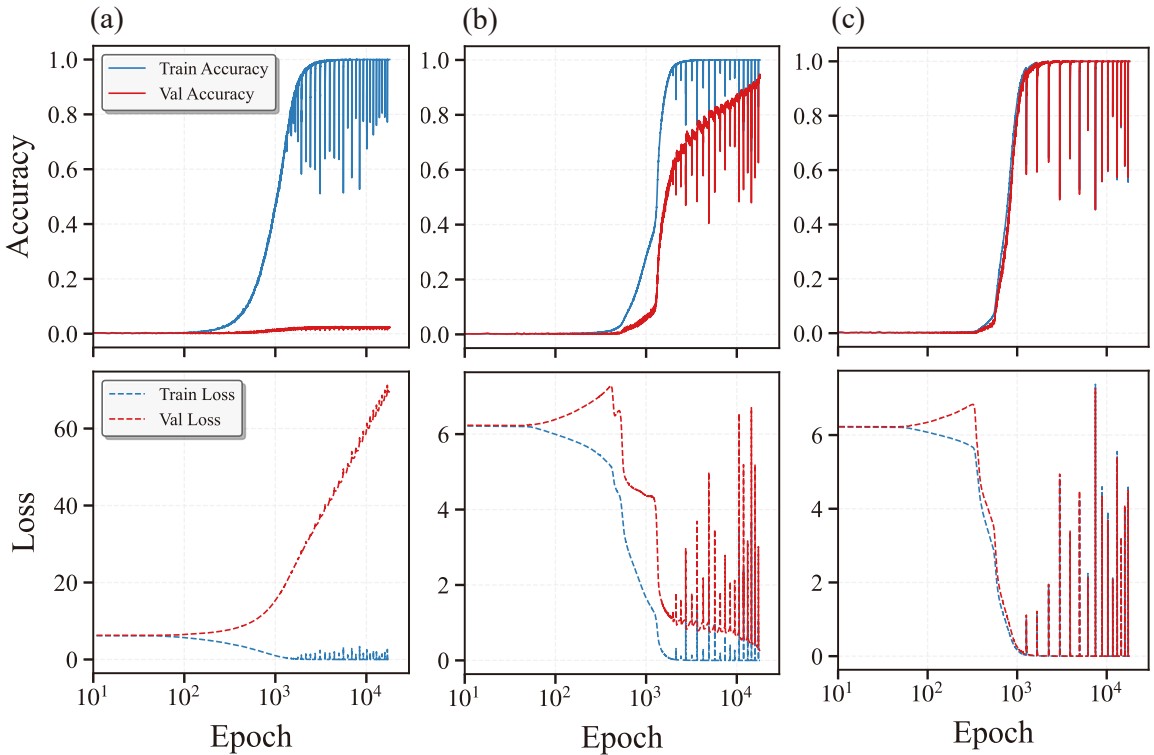

*Figure 10.* Accuracy and loss curves for three phases in the main article. (a) Memorization. (b) Semi-grokking. (c) Grokking.

### A.2. Data Generation

The following are the operations that we have tried in our work:

- $(a + b) \bmod P = c \, (P = 500)$ for $0 \le a, b < P$

- $(a + b) \bmod P = c \, (P = 499)$ for $0 \le a, b < P$

- $(a + b) \bmod P = c \, (P = 113)$ for $0 \le a, b < P$

- $(a \times b) \bmod P = c \, (P = 113)$ for $0 \le a, b < P$

For each training run, we first enumerate all possible modular arithmetic equations in the range $0 \le a, b < P$. From these, we randomly sample $\mathrm{train\_size}$ data as the training set and 2,000 as the validation set.

# B. Model and Parameter Settings

### B.1. Model for Tense-Inflection Task

In the tense-inflection task, we use an encoder-only Transformer model with the attention mask, using the following hyperparameters:

- Number of encoder layers = 4

- Number of attention heads = 8

- Hidden dimensionality = 512

- Batch size = 8.

- Loss Function: Cross Entropy Loss

- Optimizer: AdamW ($\beta_1 = 0.9$, $\beta_2 = 0.999$, $\epsilon = 1 \times 10^{-4}$)

### B.2. Settings of R2G loss strength parameter $\alpha$

Due to differences in tasks and parameter settings, users may have varying preferences for the choice of $\alpha$ in order to control the strength of R2G loss, which to encourage token separation and achieve optimal performance.

We report the best-performing $\alpha$ from all those we tried in Table 2, but due to experimental resource constraints, even better values of $\alpha$ may remain to be discovered. In the primary experiment (modular addition with $P = 500$), we set $\alpha = 0.3$. For the variants with $P = 499$ and $P = 113$, we used $\alpha = 0.27$ and $\alpha = 0.08$, respectively. In the modular multiplication experiment with $P = 113$ we used $\alpha = 0.01$, and in the tense-inflection task, we set $\alpha = 0.1$. For experiment in 2-layer decoder-only transformer, $\alpha = 0.2$.

| Task | Setting | $\alpha$ |
|---|---|---|
| Modular addition | $P = 500$ | 0.3 |
| Modular addition | $P = 499$ | 0.27 |
| Modular addition | $P = 113$ | 0.08 |
| Modular multiplication | $P = 113$ | 0.01 |
| Tense-Inflection task | – | 0.1 |
| 2-layer model | $P = 113$ | 0.2 |

*Table 2.* $\alpha$ settings for different tasks.

In our current experiments, we observed that the optimal value of $\alpha$ is intrinsically linked to the scale of the loss, the specific task complexity, and the number of tokens involved. Specifically, there is a clear positive correlation: as the number of tokens requiring spatial separation increases, a larger $\alpha$ is recommended to provide sufficient repulsive force.

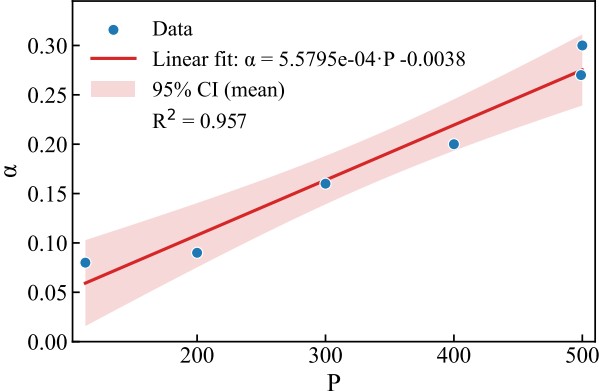

*Figure 11.* Correlation between modular base $P$ and optimal R2G coefficient $\alpha$

To further quantify this relationship, we conducted a regression analysis between the optimal $\alpha$ and the modular base $P$ (which determines the number of tokens). As shown in Figure 11, our results reveal a strong linear dependency:

$$\alpha = 5.58 \times 10^{-4} \cdot P - 3.76 \times 10^{-3}$$

. The parameter standard deviations ($\Delta k = 5.91 \times 10^{-5}, \Delta b = 2.16 \times 10^{-2}$) confirm the stability of this trend. This linear trend provides a practical rule of thumb for applying R2G to new tasks. To simplify the deployment of our method, we aim to explore adaptive mechanisms that automate the selection of $\alpha$ based on tasks in future work.

## C. Visualization

### C.1. Visualization of $Q, K, V$ Representations Vectors

Figures 12 and 13 present the 2D PCA representations of the Query ($Q$), Key ($K$), and Value ($V$) vectors for operands $b$ and $c$, respectively. The dataset we use are 2,000 validation samples, same as employed for operand $a$ in the main article. The results demonstrate that the representations for operands $a$, $b$, and $c$ are nearly identical. This indicates that the model largely disregards positional embedding differences of data.

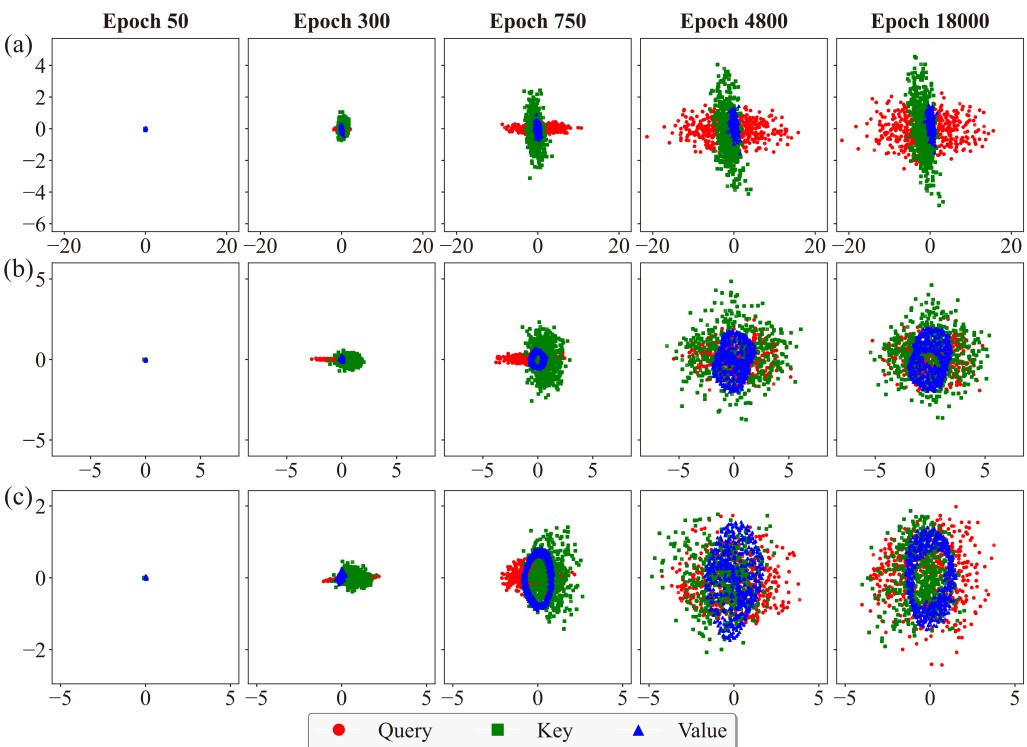

*Figure 12.* Representation of query, key, and value vector in 2D space after PCA, and five representative epochs are selected for each case. (a) memorization. (b) semi-grokking. (c) grokking. The figure exclusively illustrates the results corresponding to operand $b$.

### C.2. Value Vectors Visualization for other tokens

In this section, we visualize the representation positions of tokens 11~20 and 21~30. Consistent with the findings for tokens 1~10 presented in the main text, the results reveal distinct geometric patterns across learning phases:

- Memorization (Blue): Representations exhibit no discernible spatial organization and remain densely clustered;

- Semi-grokking (Orange): While no coherent structure emerges, representations become more dispersed;

- Grokking (Green): Merely 10 consecutive tokens form a well-defined circular arrangement with maximally separated representations for adjacent numerical tokens.

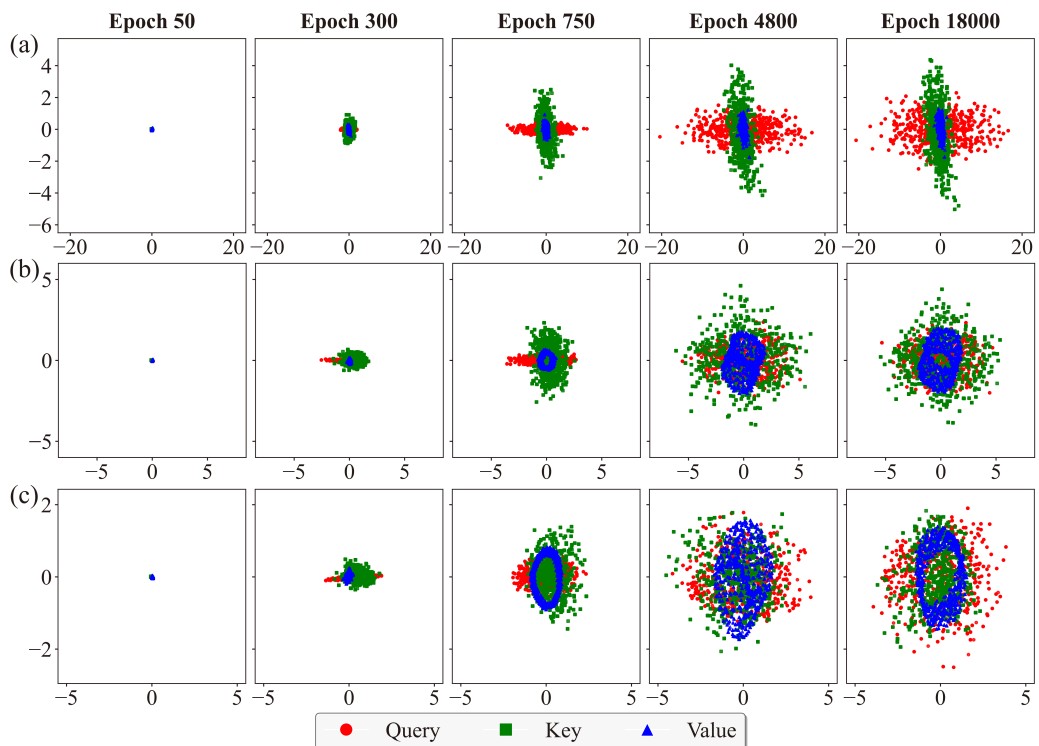

*Figure 13.* The results corresponding to operand $c$.

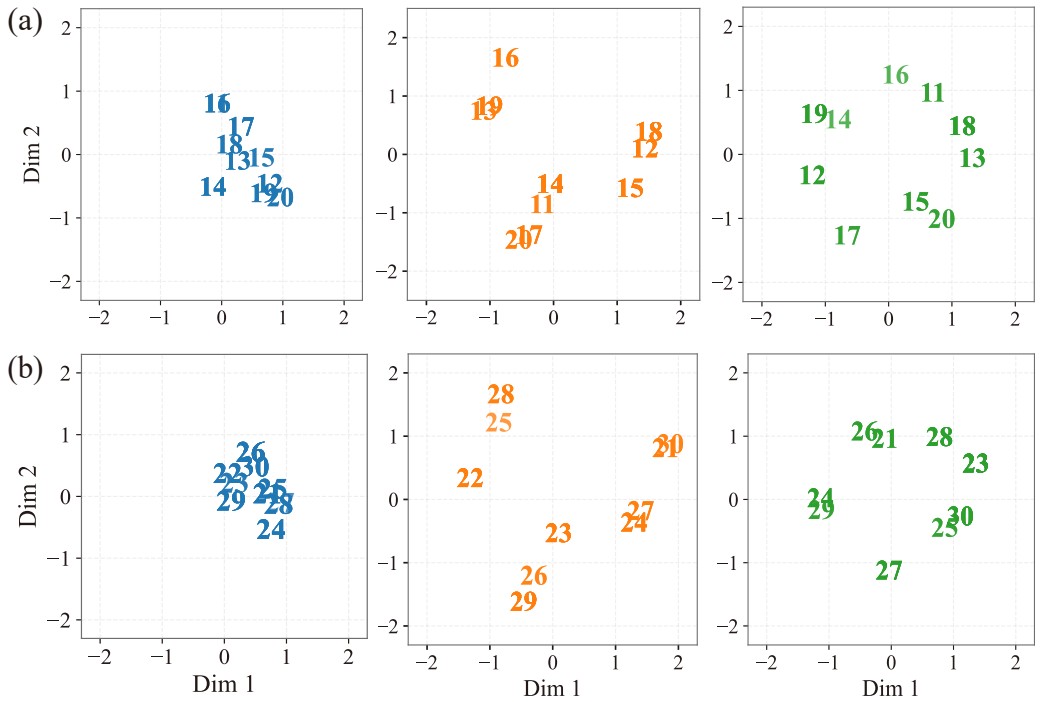

*Figure 14.* Embeddings of (a) token 11~20 and (b) 21~30 for operand $a$. Blue: memorization; Orange: semi-grokking; Green: grokking.

## C.3. Value Vectors Visualization for other operands

Figures 15 (a) and (b) present the 2D representations of tokens 1~10 for operands $b$ and $c$ after PCA, respectively. The results for operands $b$ and $c$, are the same as operand $a$ in the main article.

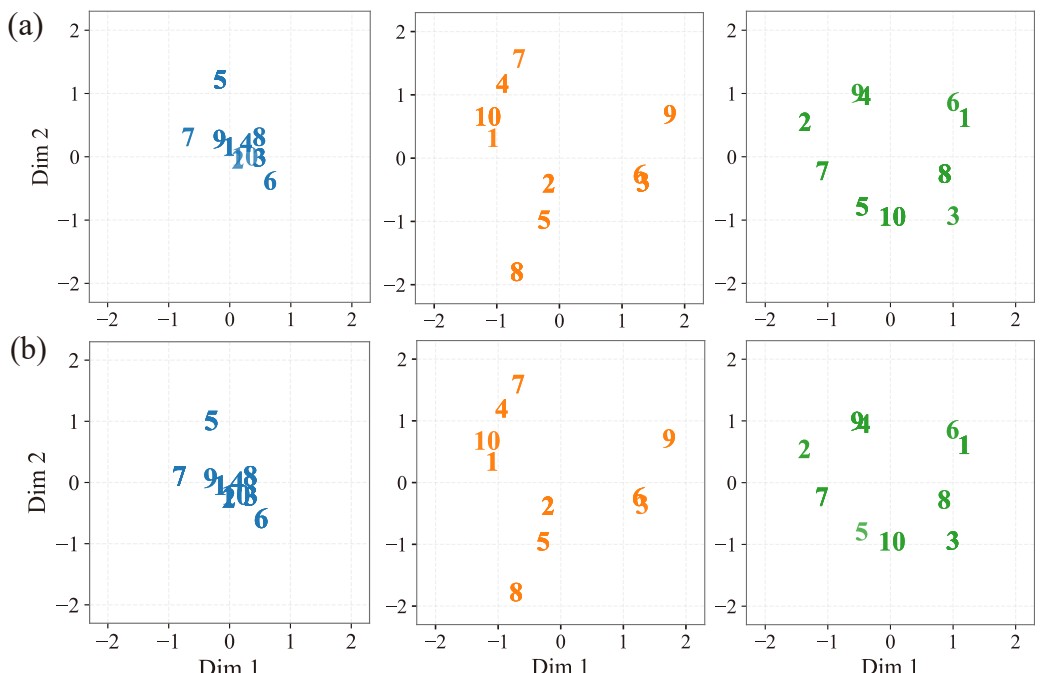

*Figure 15.* Representations of token 1~10 for (a) operand $b$ and (b) operand $c$. Blue: memorization; Orange: semi-grokking; Green: grokking.

# D. Grokking Acceleration Experiment

## D.1. Experiment Details

In each accelerating grokking experiment, we conduct 15 random seeds to report the results presented in the main text.

For the modular arithmetic tasks, we train the model for 18,000 epochs. And calculate the proportion of correct operand $c$. For the Tense-Inflection task, we train the model for 1,000,000 steps. Following previous studies (McCoy et al., 2020) (Murty et al., 2023), we measure the proportion of target verbs that are correctly inflected.

## D.2. Tense-Inflection Task

In the Tense-Inflection task the model receives an English sentence in the past tense and a tense marker (PAST or PRESENT) as input, and then output a sentence whose tense is converted according to the tense marker. When the sentence is required to be present tense, the model must determine whether each verb should take the singular or plural form. There are two rules for making this determination (McCoy et al., 2020):

(1) AGREE-RECENT: Each verb should agree with the linearly most recent noun.
(2) AGREE-SUBJECT: Each verb should agree with its hierarchically determined subject.

Although these rules yield the same predictions for sentences with linear structure, they diverge on sentences with complex hierarchical structure, such as the following sentence (a), where the correct AGREE-SUBJECT prediction is (b), while the wrong AGREE-RECENT prediction is (c):

      (a)  my zebra by the yaks swam . PRESENT
      (b)  my zebra by the yaks swims .
      (c)  my zebra by the yaks swim .

Due to the differences among tasks, we adapt a reasonable variant of the R2G loss for the tense-inflection task. Since the model has 4 layers, we apply the R2G loss to the attention module of each layer and then take the average.

In addition, consistent with prior work, the tense-inflection dataset is generated from a vocabulary with a limited number of words. Since the shortest sentence length is 3, and with the sentence-final punctuation and the tense mark added, the total length becomes 5. Therefore, we select the first five token positions as the targets for R2G loss separation. This choice aims to cover as many tokens as possible while avoiding excessive duplication, thereby applying an appropriate strength of separation to different tokens.

Following the same methodology as in the main text, we randomly sample a pair of data instances, perform separation on the tokens at the first five corresponding positions for each pair. The algorithm pseudocode is shown in Algorithm 2.

---

**Algorithm 2** R2G Loss in Tense-Inflection Task

---

1: **def R2G_Loss_PerLayer**($V$)
2:    *Input:* $V \in \mathbb{R}^{B \times L \times D}$ {value tensor of one layer}
3:    $K \leftarrow 5$ {first five token positions}
4:    **assert** $L \geq K$
5:    $perm \leftarrow \text{RandomPermutation}(B)$
6:    $total\_distance \leftarrow 0$
7: **for** $j \leftarrow 0$ **to** $K - 1$ **do**
8:     $\mathbf{v} \leftarrow V[:, j, :]$
9:     $\mathbf{v}' \leftarrow V[perm, j, :]$
10:    $d \leftarrow \|\mathbf{v} - \mathbf{v}'\|_2$
11:    $total\_distance \leftarrow total\_distance + d$
12: **end for**
13:    **return** $\frac{total\_distance}{K}$
14:
15: **def R2G_Loss_AllLayers**($\{V^{(1)}, V^{(2)}, V^{(3)}, V^{(4)}\}$)
16:    *Input:* $V^{(\ell)}$ is the value tensor for layer $\ell$
17:    $loss \leftarrow 0$
18: **for** $\ell \leftarrow 1$ **to** $4$ **do**
19:    $loss \leftarrow loss + \text{R2G\_Loss\_PerLayer}(V^{(\ell)})$
20: **end for**
21:    **return** $\frac{1}{4 \times loss}$

---

## E. Theoretical Properties of the Random Permutation Mechanism

In R2G, the repulsive loss is computed over randomly permuted pairs rather than all $\binom{B}{2}$ pairs. This raises a natural question: **does this stochastic pairing introduce optimization bias, and is the resulting gradient estimator reliable?** We answer this in two parts: we first show the estimator is unbiased and its optimization target has a well-characterized geometric attractor (Section E.2); we then bound the variance introduced by stochastic pairing and show it is not only negligible but carries a beneficial adaptive effect (Section E.3).

### E.1. Setup and Notation

Let $\mathcal{V} = \{v_1, \ldots, v_B\} \subset \mathbb{R}^d$ be a batch of Value vectors, $\pi \sim \text{Uniform}(\mathcal{S}_B)$ a uniformly random permutation, and define the single-sample distance sum $D_\pi = \sum_{i=1}^{B} \|v_i - v_{\pi(i)}\|_2$. The R2G loss for a given permutation is $\mathcal{L}_{R2G} = \lambda / D_\pi$, and the total pairwise distance sum is $\mathcal{D}_{\text{sum}} = \sum_{ij} \|v_i - v_j\|_2$.

### E.2. Unbiasedness and Geometric Attractor

**Proposition E.1** (Unbiasedness and Maximum Sum Dispersion). *For any batch $\mathcal{V}$:*

$$\mathbb{E}_\pi[D_\pi] = \frac{2}{B} \mathcal{D}_{\text{sum}}, \tag{17}$$

*so the random permutation estimator is an unbiased estimator of the total pairwise distance up to a constant factor. Consequently, minimizing $\mathbb{E}_\pi[\mathcal{L}_{R2G}]$ is equivalent in expectation to maximizing $\mathcal{D}_{\text{sum}}$—a Maximum Sum Dispersion objective whose global optimum is a uniform distribution on the hypersphere.*

*Proof.* Since $\pi(i)$ is marginally uniform over $\{1, \ldots, B\}$:

$$\mathbb{E}_\pi[D_\pi] = \sum_{i=1}^{B} \frac{1}{B} \sum_{j=1}^{B} \|v_i - v_j\|_2 = \frac{2}{B} \sum_{i<j} \|v_i - v_j\|_2 = \frac{2}{B} \mathcal{D}_{\text{sum}}. \tag{18}$$

Since $f(x) = \lambda/x$ is strictly decreasing, minimizing $\mathbb{E}[\mathcal{L}_{R2G}]$ is equivalent to maximizing $\mathbb{E}[D_\pi] \propto \mathcal{D}_{\text{sum}}$. The global optimum of Maximum Sum Dispersion under a norm constraint is a uniform arrangement on the hypersphere, matching the circular topology observed in the grokking phase (Figures 2). $\qquad \square$

The gradient estimator is similarly unbiased: under standard regularity conditions, $\mathbb{E}_\pi[\nabla_{v_k} \mathcal{L}_{R2G}] = \nabla_{v_k} \mathbb{E}_\pi[\mathcal{L}_{R2G}]$, so the optimization direction is correct on average and no systematic directional bias is introduced by the random pairing.

The global optimum of Maximum Sum Dispersion—a uniform arrangement on the hypersphere—is also the equilibrium of the classical Thomson problem (placing equal charges on a sphere to minimize electrostatic potential). This correspondence formally establishes that the attractor of R2G optimization coincides with the circular representation geometry observed in the grokking phase, providing a geometric interpretation of why R2G works.

### E.3. Variance Analysis and Beneficial Adaptive Bias

We have shown the estimator is unbiased, but each step still introduces variance from the random permutation. We now show this variance is well-controlled—and in fact carries a beneficial adaptive effect that complements the batch-level gradient scaling discussed in Section 6.2.

**Proposition E.2** (Variance bound). *Assume $\|v_i - v_j\|_2 \leq \Delta$ for all $i, j$. Then $\text{Var}(D_\pi) = \mathcal{O}(B\Delta^2)$, and the coefficient of variation satisfies:*

$$\frac{\text{Std}(D_\pi)}{\mathbb{E}[D_\pi]} = \mathcal{O}\left(B^{-1/2}\right). \tag{19}$$

*Proof.* Write $D_\pi = \sum_{i=1}^{B} X_i$ where $X_i = \|v_i - v_{\pi(i)}\|_2$. By assumption, $X_i \in [0, \Delta]$ for all $i$. The variance of $D_\pi$ expands as:

$$\text{Var}(D_\pi) = \sum_{i=1}^{B} \text{Var}(X_i) + \sum_{i \neq j} \text{Cov}(X_i, X_j). \tag{20}$$

**Bounding the diagonal terms.** Since $X_i \in [0, \Delta]$, Popoviciu's inequality gives $\text{Var}(X_i) \leq \Delta^2/4$ for each $i$, so:

$$\sum_{i=1}^{B} \text{Var}(X_i) \leq \frac{B\Delta^2}{4}. \tag{21}$$

**Bounding the off-diagonal terms.** For $i \neq j$, we compute $\text{Cov}(X_i, X_j) = \mathbb{E}[X_i X_j] - \mathbb{E}[X_i]\mathbb{E}[X_j]$. Since $\pi(i)$ is marginally uniform over $\{1, \ldots, B\}$:

$$\mathbb{E}[X_i] = \frac{1}{B} \sum_{k=1}^{B} \|v_i - v_k\|_2. \tag{22}$$

For $i \neq j$, the pair $(\pi(i), \pi(j))$ is uniform over all ordered pairs with $\pi(i) \neq \pi(j)$, so:

$$\mathbb{E}[X_i X_j] = \frac{1}{B(B-1)} \sum_{k \neq l} \|v_i - v_k\|_2 \cdot \|v_j - v_l\|_2. \tag{23}$$

Meanwhile:

$$\mathbb{E}[X_i]\mathbb{E}[X_j] = \frac{1}{B^2} \sum_{k=1}^{B} \sum_{l=1}^{B} \|v_i - v_k\|_2 \cdot \|v_j - v_l\|_2. \tag{24}$$

The difference between these two expressions arises solely from the $B$ diagonal terms $k = l$ that appear in $\mathbb{E}[X_i]\mathbb{E}[X_j]$ but not in $\mathbb{E}[X_i X_j]$. Each such term is bounded by $\Delta^2$, and a straightforward calculation gives:

$$|\mathrm{Cov}(X_i, X_j)| = \mathcal{O}\left(\frac{\Delta^2}{B}\right). \tag{25}$$

Summing over all $B(B-1)$ off-diagonal pairs:

$$\left|\sum_{i \neq j} \mathrm{Cov}(X_i, X_j)\right| \leq B(B-1) \cdot \mathcal{O}\left(\frac{\Delta^2}{B}\right) = \mathcal{O}(B\Delta^2). \tag{26}$$

**Combining.** Both the diagonal and off-diagonal contributions are $\mathcal{O}(B\Delta^2)$, giving $\mathrm{Var}(D_\pi) = \mathcal{O}(B\Delta^2)$, and hence $\mathrm{Std}(D_\pi) = \mathcal{O}(\sqrt{B}\,\Delta)$.

**Coefficient of variation.** From Proposition E.1, $\mathbb{E}[D_\pi] = \frac{2}{B}\mathcal{D}_{\mathrm{sum}}$. Since there are $\binom{B}{2}$ pairwise distances each of average value $\bar{d}$, we have $\mathcal{D}_{\mathrm{sum}} = \Omega(B^2\bar{d})$, so $\mathbb{E}[D_\pi] = \Omega(B\bar{d})$. Therefore:

$$\frac{\mathrm{Std}(D_\pi)}{\mathbb{E}[D_\pi]} = \frac{\mathcal{O}(\sqrt{B}\,\Delta)}{\Omega(B\bar{d})} = \mathcal{O}\left(\frac{1}{\sqrt{B}}\right). \tag{27}$$

$\square$

As representations disperse during training, pairwise distances become more uniform across permutations, so $\sigma_D^2$ decreases as training progresses. For any smooth function $g(D)$ of a random variable $D$ with mean $\bar{D}$, a first-order Taylor expansion gives $\mathrm{Var}(g(D)) \approx [g'(\bar{D})]^2 \cdot \mathrm{Var}(D)$. Here $g(D) = \lambda/D$, so $g'(D) = -\lambda/D^2$, and:

$$\mathrm{Var}(\mathcal{L}_{R2G}) \approx \left(\frac{\lambda}{\bar{D}^2}\right)^2 \mathrm{Var}(D_\pi) = \frac{\lambda^2}{\bar{D}^4} \cdot \mathrm{Var}(D_\pi). \tag{28}$$

To bound $\bar{D}$ from below, recall from Proposition E.1 that $\bar{D} = \frac{2}{B}\mathcal{D}_{\mathrm{sum}}$. Since $\mathcal{D}_{\mathrm{sum}} = \sum_{i<j}\|v_i - v_j\|$ contains $\binom{B}{2} = \frac{B(B-1)}{2}$ terms each of average value $\bar{d}$, we have $\mathcal{D}_{\mathrm{sum}} = \frac{B(B-1)}{2}\bar{d}$, and therefore:

$$\bar{D} = \frac{2}{B} \cdot \frac{B(B-1)}{2}\bar{d} = (B-1)\bar{d} = \Omega(B\bar{d}). \tag{29}$$

Substituting $\mathrm{Var}(D_\pi) = \mathcal{O}(B\Delta^2)$ and $\bar{D} = \Omega(B\bar{d})$:

$$\mathrm{Var}(\mathcal{L}_{R2G}) = \mathcal{O}\left(\frac{\lambda^2 B\Delta^2}{B^4\bar{d}^4}\right) = \mathcal{O}\left(\frac{\lambda^2\Delta^2}{B^3\bar{d}^4}\right), \tag{30}$$

decaying cubically in batch size. By Jensen's inequality applied to the convex function $1/D$:

$$\mathbb{E}\left[\frac{\lambda}{D_\pi}\right] \geq \frac{\lambda}{\mathbb{E}[D_\pi]} = \frac{\lambda}{\bar{D}} = \Omega\left(\frac{\lambda}{B\bar{d}}\right) = \Omega(\lambda B^{-1}\bar{d}^{-1}), \tag{31}$$

so the expected loss is lower-bounded by $\Omega(\lambda B^{-1}\bar{d}^{-1})$. In our experiments we use full-batch training, so $B$ equals the full training set size. The coefficient of variation therefore satisfies:

$$\frac{\mathrm{Std}(\mathcal{L}_{R2G})}{\mathbb{E}[\mathcal{L}_{R2G}]} \leq \frac{\mathcal{O}(\lambda\Delta B^{-3/2}\bar{d}^{-2})}{\Omega(\lambda B^{-1}\bar{d}^{-1})} = \mathcal{O}\left(\frac{\Delta}{B^{1/2}\bar{d}}\right) \to 0 \quad \text{as } B \to \infty, \tag{32}$$

confirming that permutation noise is provably negligible in our training setting.

Beyond negligibility, the residual variance carries a beneficial effect. Since $f(x) = 1/x$ is strictly convex, Jensen's inequality gives $\mathbb{E}[\lambda/D_\pi] \geq \lambda/\mathbb{E}[D_\pi]$. A second-order Taylor expansion around $\bar{D} = \mathbb{E}[D_\pi]$ makes this gap explicit:

$$\mathbb{E}\left[\frac{\lambda}{D_\pi}\right] \approx \frac{\lambda}{\bar{D}} + \frac{\lambda\sigma_D^2}{\bar{D}^3}. \tag{33}$$

The correction term $\lambda\sigma_D^2/\bar{D}^3$ depends on both $\sigma_D^2$ and $\bar{D}$. In the memorization phase, representations are collapsed: $\bar{D}$ is small and pairwise distances are highly heterogeneous across permutations, so $\sigma_D^2$ is large. The correction term is therefore large, meaning the effective repulsive strength is automatically amplified. In the grokking phase, representations are already well-dispersed: $\bar{D}$ is large and distances are uniform across permutations, so $\sigma_D^2$ is small and the correction vanishes. In other words, the stochastic permutation mechanism naturally strengthens the repulsion when structural reorganization is most needed and relaxes it once the geometry has formed—a self-regulating property that requires no task-specific tuning and is a direct consequence of the reciprocal form of $\mathcal{L}_{R2G}$.

## F. Causal Evidence

The mechanistic analysis in Section 4 establishes a consistent *correlation* between V-vector geometric reorganization and the onset of grokking. To probe the causal direction of this relationship, we provide three levels of evidence following the framework of Pearl's Ladder of Causation (Pearl, 2018).

**Level 1: Association** $P(Y|X)$

Our mechanistic analysis (Figures 2, 3, 4) documents a consistent correlation between V-vector geometric reorganization and the onset of grokking across training phases: the Pearson correlation between numerical token differences and embedding distances grows from near-zero in memorization to a concentrated distribution in grokking, establishing the observational foundation.

**Level 2: Intervention** $P(Y|\mathbf{do}(X))$

R2G Loss itself constitutes a controlled do-operator intervention. Under strictly identical data, model, and optimizer conditions, R2G directly manipulates the hidden states geometry and observes a corresponding change in generalization behavior. If geometric reorganization were merely a by-product of grokking rather than its cause, actively enforcing it should not induce grokking—yet it consistently does, falsifying the null hypothesis.

**Level 3: Counterfactual disruption** $P(Y_x|X', Y')$

To further probe the causal direction, we designed a geometric disruption experiment. Starting from runs already in the grokking phase, we injected additive Gaussian perturbations $\epsilon \sim \mathcal{N}(0, \sigma^2 I)$ into all token hidden states at different training epochs (400, 600, 800, 1000), with noise scales $\sigma \in \{0.1, 1.0, 2.0\}$, and monitored whether validation accuracy recovered within a fixed number of additional training steps.

*Table 3.* Recovery rate of validation accuracy after geometric disruption. "Before/after rise" refers to whether the perturbation occurs before or after the sharp increase in validation accuracy.

| Noise Scale $\sigma$ | Timing | Recovery Rate |
|:---:|:---:|:---:|
| 0.1 | before rise | 60% |
| 0.1 | after rise | 100% |
| 1.0 | before rise | 0% |
| 1.0 | after rise | 80% |
| 2.0 | before rise | 0% |
| 2.0 | after rise | 0% |

The results in Table 3 reveal a clear causal signature: the timing of disruption relative to the validation accuracy rise proves decisive: geometric structure becomes progressively more resistant to perturbation as training advances, and models closer to full grokking are better able to withstand and self-repair from identical perturbation strengths. This monotonic consolidation of geometric structure, which stabilizes before the sharp rise in validation accuracy, establishes a clear temporal precedence: the formation of geometric structure precedes, and causally drives, the emergence of generalization.

Together, these three levels of evidence: observation, intervention, and counterfactual disruption, provide a principled causal account consistent with the do-calculus framework.

## G. Robustness to Weight Decay

To determine whether R2G provides a geometric benefit independent of weight decay regularization, we swept $\gamma \in \{0, 0.001, 0.01, 0.1\}$ and evaluated both the baseline and R2G at each value on the $P = 500$ modular addition task with dataset size 26,650.

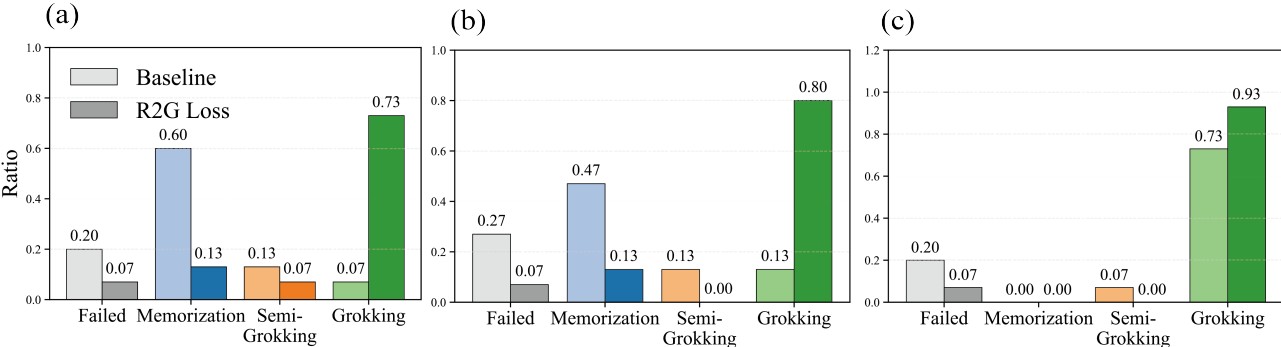

*Figure 16.* Effect of R2G loss under different weight decay settings. Experiments conducted on modular addition task (P=500) with datasize 26650. (a) weight decay=0, (b) weight decay=0.001, (c) weight decay=0.1.

As shown in Figure 16, R2G consistently improves grokking rates across all weight decay settings, including the no-weight-decay case ($\gamma = 0$). These results demonstrate that R2G provides an *independent geometric benefit* that is additive to, rather than reliant on, weight decay regularization. This is consistent with our theoretical analysis: R2G operates directly on the angular geometry of V-vector hidden states, a mechanism orthogonal to the norm-shrinking effect of weight decay.

## H. Ablation Study

### H.1. Ablation Study on Decoupled Variants

This appendix provides the algorithmic details of the decoupled variants of the R2G loss used in the ablation studies, including the Norm-only loss ($L_N$) and the Angular-only loss ($L_A$). These variants are designed to selectively isolate the effects of radial and angular components in representation updates, as discussed in the main text.

Algorithms 3 and 4 present the full training procedures for $L_N$ and $L_A$, respectively. Both algorithms closely follow the structure of the original R2G loss, differing only in the geometric component they act upon. In particular, the Norm-only variant operates exclusively on magnitudes, while the Angular-only variant normalizes representations onto the hypersphere and applies repulsion based on angular separation.

Importantly, these decoupled losses preserve the simplicity and computational efficiency of R2G. They introduce no additional model parameters and incur negligible overhead beyond standard embedding operations, ensuring that observed

---

**Algorithm 3** Norm-only Loss

---

**Input:** $\mathbf{V} \in \mathbb{R}^{B \times L \times D}$, Position set $\mathcal{P}$, Scaling factor $\lambda$
**Output:** Norm-only loss $\mathcal{L}_N$
$S_{total} \leftarrow 0$
**for** each position $p \in \mathcal{P}$ **do**
  $\mathbf{v}_p \leftarrow \mathbf{V}[:, p, :]$
  $\mathbf{n}_{p,i} \leftarrow \|\mathbf{v}_{p,i}\|_2$   for each $i \in \{1, \ldots, B\}$
  $s_p \leftarrow \sum_{i=1}^{B} \mathbf{n}_{p,i}$
  $S_{total} \leftarrow S_{total} + s_p$
**end for**
$\mathcal{L}_N \leftarrow \dfrac{\lambda}{S_{total}}$
**return** $\mathcal{L}_N$

---

---

**Algorithm 4** Angular-only Loss

---

**Input:** $\mathbf{V} \in \mathbb{R}^{B \times L \times D}$, Position set $\mathcal{P}$, Scaling factor $\lambda$
**Output:** Angular-only loss $\mathcal{L}_A$
$D_{total} \leftarrow 0$
**for** each position $p \in \mathcal{P}$ **do**
    $\mathbf{v}_p \leftarrow \mathbf{V}[:, p, :]$
    $\hat{\mathbf{v}}_p \leftarrow \dfrac{\mathbf{v}_p}{\|\mathbf{v}_p\|_2}$
    $\pi \leftarrow \text{RandomPermutation}(B)$
    $\mathbf{v}_{p,rand} \leftarrow \hat{\mathbf{v}}_p[\pi, :]$
    $d_p \leftarrow \sum_{i=1}^{B} \|\hat{\mathbf{v}}_{p,i} - \mathbf{v}_{p,rand,i}\|_2$
    $D_{total} \leftarrow D_{total} + d_p$
**end for**
$\mathcal{L}_A \leftarrow \dfrac{\lambda}{D_{total}}$
**return** $\mathcal{L}_A$

---

differences in grokking behavior can be attributed to the targeted geometric interventions rather than to implementation complexity.

### H.2. Ablation Study on Formulation Variants

This section details the algorithmic implementations of the two normalization-based variants evaluated in Section 7.2, and provides the full experimental results.

**LayerNorm** normalizes each Value vector over the feature dimension to zero mean and unit variance before applying quadratic repulsion, preserving relative geometric structure while controlling absolute scale. **Hyperball** projects all Value vectors onto the unit hypersphere via $\ell_2$-normalization before applying quadratic repulsion, discarding all magnitude information and operating purely on angular geometry; this is equivalent to our Angular-only ablation under a quadratic rather than reciprocal objective.

As shown in Figure 17, both normalization-based variants underperform R2G. Although LayerNorm controls the absolute scale of representations, its quadratic loss gradient scales as $\mathcal{O}(\sin\theta_{kk'})$, which vanishes as $\theta \to 0$ when representations are collapsed, providing insufficient repulsion precisely at the memorization phase where structural intervention is most critical. The Hyperball variant achieves only 33%, much lower than R2G Loss form.

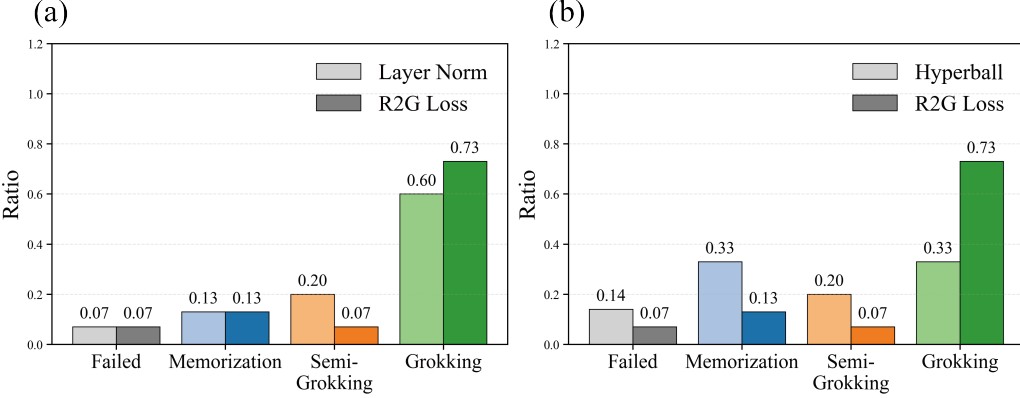

*Figure 17.* Comparison of grokking success rates under alternative loss formulations on the $P = 500$ modular addition task. (a) Quadratic loss with LayerNorm normalization. (b) Quadratic loss with Hyperball ($\ell_2$) normalization.

---

**Algorithm 5** LayerNorm Variant

---

**Input:** $\mathbf{V} \in \mathbb{R}^{B \times L \times D}$, Position set $\mathcal{P} = \{0, 2, 6\}$, Scaling factor $\lambda$
**Output:** LayerNorm loss $\mathcal{L}_{LN}$
$S_{total} \leftarrow 0$
**for** each position $p \in \mathcal{P}$ **do**
    $\mathbf{v}_p \leftarrow \mathbf{V}[:, p, :]$
    $\hat{\mathbf{v}}_p \leftarrow \mathrm{LayerNorm}(\mathbf{v}_p)$     {normalize over feature dim: mean 0, var 1}
    $\pi \leftarrow \mathrm{RandomPermutation}(B)$
    $\mathbf{d}_p \leftarrow \|\hat{\mathbf{v}}_p - \hat{\mathbf{v}}_p[\pi, :]\|_2$
    $S_{total} \leftarrow S_{total} + \sum_{i=1}^{B} (\mathbf{d}_p)_i^2$
**end for**
$\mathcal{L}_{LN} \leftarrow -S_{total} \,/\, \lambda$
**return** $\mathcal{L}_{LN}$

---

---

**Algorithm 6** Hyperball Variant

---

**Input:** $\mathbf{V} \in \mathbb{R}^{B \times L \times D}$, Position set $\mathcal{P} = \{0, 2, 6\}$, Scaling factor $\lambda$
**Output:** Hyperball loss $\mathcal{L}_{HB}$
$S_{total} \leftarrow 0$
**for** each position $p \in \mathcal{P}$ **do**
    $\mathbf{v}_p \leftarrow \mathbf{V}[:, p, :]$
    $\hat{\mathbf{v}}_p \leftarrow \mathbf{v}_p \,/\, \|\mathbf{v}_p\|_2$     {$\ell_2$-normalize onto unit hypersphere}
    $\pi \leftarrow \mathrm{RandomPermutation}(B)$
    $\mathbf{d}_p \leftarrow \|\hat{\mathbf{v}}_p - \hat{\mathbf{v}}_p[\pi, :]\|_2$
    $S_{total} \leftarrow S_{total} + \sum_{i=1}^{B} (\mathbf{d}_p)_i^2$
**end for**
$\mathcal{L}_{HB} \leftarrow -S_{total} \,/\, \lambda$
**return** $\mathcal{L}_{HB}$

---

# I. Preliminary Analysis on Larger Pretrained Models

To assess whether V-vector angular reorganization is a general phenomenon beyond shallow Transformers, we performed a preliminary analysis of OLMo-2-1B (OLMo et al., 2025) across publicly available pretraining checkpoints (9 checkpoints selected from step 0 to step 1M).

## I.1. Numerical Structure Emergence

We extracted V-vectors for number tokens (0–99) across all 16 layers and computed the Pearson correlation $r$ between numerical token differences $|i - j|$ and V-vector Euclidean distances, analogous to the analysis in Figure 4 of the main paper. As shown in Figure 18(a), the correlation increases monotonically in Layer 0:

$$r = -0.006 \text{ (step 0)} \;\longrightarrow\; r = 0.453 \text{ (step 50K)} \;\longrightarrow\; r = 0.750 \text{ (step 1M)}.$$

The remaining 15 layers exhibit the same qualitative trend (Figure 18(b)), confirming that numerical relationships are progressively encoded into V-vector geometry during large-scale pretraining—consistent with our mechanistic findings on small Transformers.

## I.2. Semantic Structure Emergence

We further examined whether semantic relationships are encoded in V-vector geometry by observing tokens of three categories (*animals*, *colors*, *numbers*) and computing the intra-/inter-group distance ratio $\rho = \bar{d}_{\mathrm{intra}}/\bar{d}_{\mathrm{inter}}$. As shown in Figure 18(c), $\rho$ decreases monotonically from approximately 1.0 at random initialization, indicating that semantically related tokens are progressively clustered into the same region of V-space as pretraining proceeds.

## I.3. Discussion

Together, these analyses suggest that V-vector structural reorganization is not an artifact of shallow architectures or small algorithmic datasets, but a general phenomenon observable in large-scale pretraining. This provides promising potential for broader downstream applications of R2G-style geometric interventions, and we regard a more systematic investigation of this phenomenon as a valuable direction for future work.

**Limitations.** This analysis is preliminary: (1) we examine only 9 checkpoints and a single model family; (2) the causal relationship between V-vector geometry and downstream task performance in LLMs remains to be established; (3) the connection to grokking-like phenomena in large models requires further investigation. We include these results as supporting evidence for the generality of our mechanistic findings rather than as a definitive empirical claim.

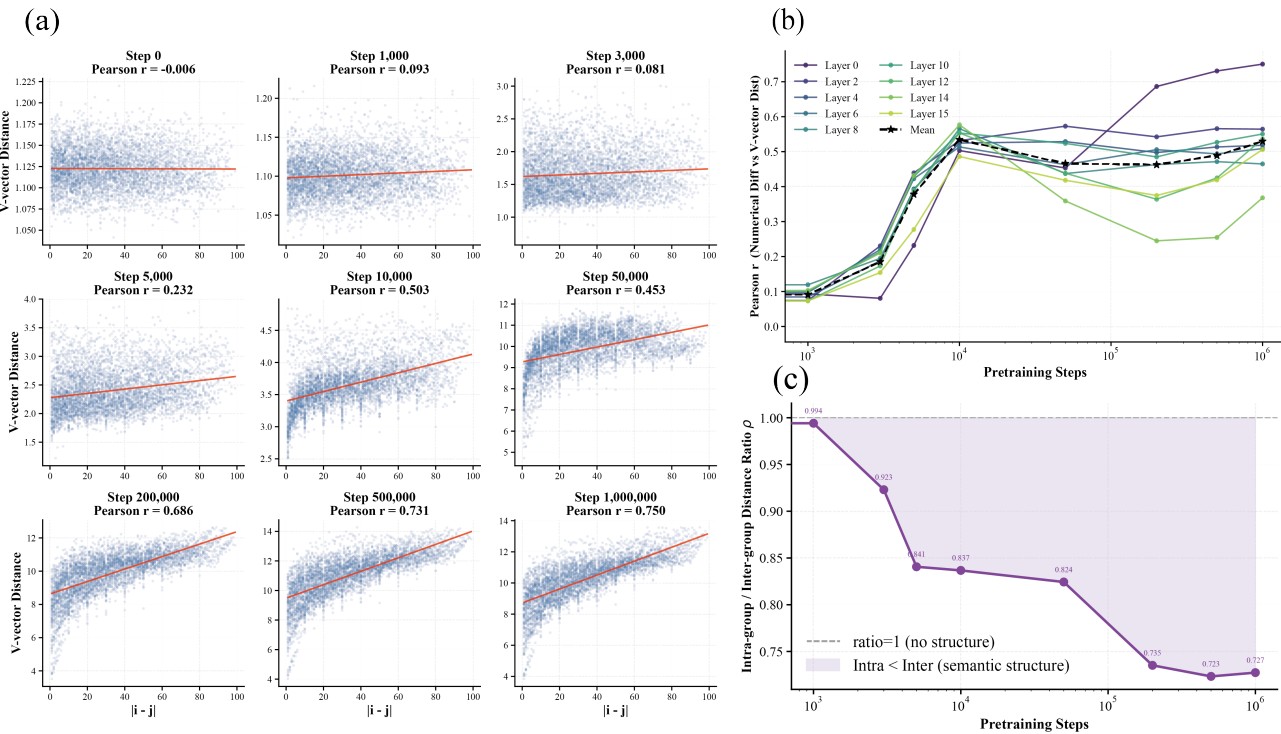

*Figure 18.* Preliminary analysis of $V$ vectors on pretrained models OLMo. (a) The relationship between the distance of $V_i$ and $V_j$ in $V$ space (geometric distance) and the difference $|i - j|$ (numerical distance) emerges during training.(b) Analysis results across all 16 layers, where the vertical axis represents the Pearson correlation coefficient from (a), and the horizontal axis represents training steps (c) Semantic experiment, where the vertical axis shows ratio $\rho = \mathrm{intra}_{\mathrm{mean}}/\mathrm{inter}_{\mathrm{mean}}$, and the horizontal axis shows training steps.

