# OpenReview forum: "The Geometric Origin of Grokking: Accelerating Generalization via Active Structural Reorganization"
_ICML.cc/2026/Conference — ICML 2026 regular_

### Official Review · Reviewer_Ep2F · 2026-03-06

**Soundness:** 3
**Presentation:** 3
**Significance:** 3
**Originality:** 3
**Overall Recommendation:** 5
**Confidence:** 5

**Summary:**

This paper argues that grokking arises from a geometric reorganization of Transformer representations and proposes an auxiliary R2G (Repel-to-Grokking) loss to accelerate this reorganization phase. Experiments on modular arithmetic and a syntactic task show that R2G loss, by encouraging embedding separation, increases the fraction of runs that reach grokking (and can reduce time-to-grokking when it does).

**Compliance With Llm Reviewing Policy:**

Affirmed.

**Final Justification:**

I maintain my accept score. I continue to find the paper technically solid, well presented, and original in linking a geometric account of grokking to a simple, practical intervention. The embedding-geometry analysis and targeted ablations make R2G well motivated and useful.

**Key Questions For Authors:**

Have the authors compared R2G to established grokking acceleration approaches such as Grokfast under matched compute budgets and comparable hyperparameter tuning effort? If so, how does R2G compare in terms of grokking success rate and time-to-generalization? While the primary contribution of this work is mechanistic and geometric rather than purely acceleration-focused, a brief comparison or discussion, even in an appendix, would help situate R2G more clearly within the existing acceleration literature.

**Limitations:**

Yes.

**Strengths And Weaknesses:**

**Soundness:** The work is technically solid. The claims regarding the geometrical cause of grokking are supported by detailed mechanistic experiments and some nice theoretical discussions. Overall, this paper  meets the ICML bar (see core strengths below), and would be a good addition to the literature, with some minor notes taken into consideration  (see core weaknesses)

**Presentation:** The paper is generally clear both in terms of writing style and figures used. One minor concern is that the bullet points around “failed/memorization/semi-grokking/grokking” feel somewhat repetitive and could be streamlined into a single unified definition block or table of observations.

**Originality:** The paper offers an original and relevant contribution: a mechanistic geometric account of grokking paired with an “active” intervention (R2G) derived from that analysis. This is a fresh angle that manages to distinguish itself from substantial prior work on grokking (see Core Strengths below).

**Significance:** The paper is significant because it provides a lightweight, plug-and-play loss for the reader to use in their experiments. However, stronger baseline comparisons would further strengthen its projected significance (see core weaknesses below).

**Core Strengths:**
1. _Mechanistic evidence linking grokking to embedding vector geometry._ The paper goes beyond performance-improvement curves to provide supporting evidence for its mechanistic claim through embedding visualisation (Figure 2; 3) and analysis of distance distributions (Figure 4). This provides a highly motivated context for the introduced R2G loss algorithm, which is modular enough to be added into the standard cross entropy loss signal.

2. _Concrete performance improvements._  The empirical evaluation goes beyond showing a faster training curve in a single setting. In the main modular-addition task (Figure 1), R2G changes the phase trajectory of training. Runs that would otherwise remain in memorization or semi-grokking transition to full grokking under the same data and compute budget. This effect is quantified in Section 5.2 and Figures 6 & 7, which report a substantial increase in the fraction of runs that reach grokking across settings. Moreover, the targeted ablation in Figure 8 shows that applying R2G to Value vectors is more effective than applying it to Q or K, directly supporting the paper’s hypothesis.

3. _Theoretical explanation that aligns with the empirical claim (angular vs norm)._ The radial/angle decomposition provides a principled rationale for why grokking might be gated by slow angular reorganization while norms can change quickly without restructuring representations. This is a nice “theory layer” that strengthens the otherwise empirical geometric story.

4. _Practical simplicity and reproducibility (easy for others to build on)._ R2G is a modular loss term, and is clearly specified, making it straightforward to re-implement and test. This has scientific value as the reader can attempt to reproduce/validate the claims, run comparisons, or try the method in their own grokking settings.

**Core Weaknesses:**
1. _Cross-architecture generality._ Grokking has now been observed across a broader range of models and depths beyond two-layer Transformers [1,2]. It would therefore strengthen the paper to test whether the R2G loss extends to other architectures or training regimes. That said, the controlled two-layer setting is consistent with prior acceleration work such as Grokfast [3], and there is no strong a priori reason to believe that R2G should be architecture-specific, since it operates on intermediate Value representations that are common across many decoder-only Transformer variants.

2. _Sample-size axis vs time axis tension._ The paper studies phase behavior primarily by sweeping dataset size (Section 3.3; Figure 2), whereas grokking is typically characterized as a training-time phenomenon. This is a deliberate design choice used to expose distinct internal regimes in a controlled way. However, despite being introduced along a non-time axis, the authors do a good job of reconnecting these observations to the time dimension, both experimentally and theoretically, by showing that R2G materially accelerates the onset of generalization.

3. _Use of the term “embedding”._ One small issue is that the manuscript uses the term “embedding” for intermediate representations beyond the input embedding layer (e.g., Fig. 2; App. C.1; Fig. 5; Alg. 1; Sec. 6). Reserving “token embedding” for the input layer and using “hidden states” or “representations” for intermediate activations would improve clarity, particularly in the multi-layer experiments, as this would be a more standard use of terminology.



**References**

[1] Humayun et al., “Deep Neural Networks Always Grok and Here is Why”, ICML 2024.

[2] Li, et al. “Grokking in LLM Pretraining? Monitor Memorization-to-Generalization without Test.”, 2025.

[3] Lee et al., “Grokfast: Accelerated Grokking by Amplifying Slow Gradients,” 2024.

---

> ### Author Rebuttal · Authors · 2026-03-30
>
> We thank you for the careful and generous reading of our work, and for the detailed, constructive feedback.
>
> ---
>
> **Response to Core Weakness (1): Cross-architecture generality**
>
> We thank you for the thoughtful observation that R2G's operation on Value representations provides a principled basis for architectural generality.
>
> We would like to highlight that our experiments already span two architecturally distinct settings: the modular arithmetic tasks use a 1-layer decoder-only Transformer ($d_\text{model}=48$), while the tense-inflection task uses a 4-layer encoder-only Transformer ($d_\text{model}=512$, 8 attention heads, Appendix B.1). The consistent effectiveness of R2G across these two settings, which differ in depth, width, architecture type, and task domain.
>
> Furthermore, to address the concern about model depth, we have conducted additional experiments using a **2-layer decoder-only Transformer** ($d_\text{model}=128$) on the $P=500$ modular addition task. R2G loss achieves a **73%** grokking rate compared to 7% for the baseline, confirming that the acceleration effect persists as model depth increases. We will incorporate these results into the revision.
>
> We agree that broader evaluation across more architectures and training regimes remains a valuable direction, and we will explicitly discuss this as future work in the revised manuscript.
>
> ---
>
> **Response to Core Weakness (2): Sample-size axis vs. time axis tension**
>
> The use of dataset size as the primary axis for phase characterization is indeed a deliberate methodological choice. Training-time analysis alone conflates two distinct sources of variance: the stochasticity of optimization trajectories and the intrinsic generalization capacity of the model given a fixed data regime. By fixing training duration and sweeping dataset size, we isolate the latter, exposing structurally distinct internal regimes (memorization, semi-grokking, grokking) in a controlled and reproducible manner. This design is consistent with Huang et al. (2024), who adopt the same phase taxonomy.
>
> ---
>
> **Response to Key Question: Comparison with Grokfast and NeuralGrok**
>
> We have conducted additional experiments comparing R2G against Grokfast and NeuralGrok under identical training datasets on the modular addition task ($P = 500$, dataset size = $26,650$).
>
> - **Grokfast** applies a low-pass filter (EMA) to the gradient stream to amplify slow, low-frequency components that drive generalization; we tuned its cutoff coefficient $\alpha$ to the task-optimal value of $0.3$.
>
> - **NeuralGrok** trains an auxiliary neural amplifier that adaptively re-weights gradients; we held out a portion of training data for the amplifier and tuned its scale parameter $c_{\text{norm}} = 0.2$ and update frequency $\text{inner loop steps} = 2$ for optimal performance.
>
> Under the same dataset, the results are as follows:
>
> | Method | Grokking Gain |
> |---|---|
> | R2G | **73%** |
> | Grokfast | 60% |
> | NeuralGrok | 27% |
>
> Moreover, among runs that successfully reach grokking, R2G converges the fastest. Visualization results (phase-distribution bar chart and epoch-wise grokking-rate curve) are provided in **https://anonymous.4open.science/r/anonymous-figure-icml/figures.pdf - Fig 1**.
>
> These results highlight a qualitative distinction: Grokfast and NeuralGrok focus on temporal acceleration of gradient signals, whereas R2G directly restructures the embedding manifold, making it a complementary and more geometrically principled intervention for promoting phase transitions.
>
> We will include this comparison in the revised manuscript.
>
> ---
>
> **Response to Weakness (3):**
>
> Finally, we will address all minor issues (terminology: "token embedding" vs. "hidden states/representations", and the streamlining of the phase definition blocks) in the revised manuscript.
>
> ---
>
> We hope these additional results and clarifications address your concerns. We thank you again for the constructive and fair assessment, and look forward to incorporating these improvements into the revised manuscript.

---

> > ### Author Rebuttal · Reviewer_Ep2F · 2026-03-31
> >
> > I appreciate the comprehensive rebuttal and maintain my accept score.

---

### Official Review · Reviewer_9CR9 · 2026-03-11

**Soundness:** 3
**Presentation:** 3
**Significance:** 1
**Originality:** 1
**Overall Recommendation:** 2
**Confidence:** 4

**Summary:**

The authors propose a new regularization technique to improve grokking. This involves maximizing angular separations which steers the model's internal representations towards ones suitable for grokking. They then perform ablation studies to clearly delineate the role of norm and angular separations.

**Compliance With Llm Reviewing Policy:**

Affirmed.

**Final Justification:**

I appreciate the prompt response from the authors, but the settings that I suggested need more careful tuning and experimentation. I thus maintain my rating.

**Key Questions For Authors:**

(1) The R2G loss needs better motivation, for example they are effectively implementing a (inverse) MSE loss, while using an CSE for the actual logits. Is the reciprocal nature of the loss necessary? What if we just have $-\sum_{i=1}^n || v_i - v_i'||^2$ ?

(2) The theoretical analysis performed is elementary. A better exposition could be to discuss possible biases introduced by the R2G loss (and in particular, the randomness introduced through a random permutation of the vectors).

**Limitations:**

No further limitations need to be discussed.

**Strengths And Weaknesses:**

Strengths:

(1) The paper presents an interesting extension of the grokking literature by introducing a new loss function. This loss function has a clear and interpretable role, which is consistent with different observations made by the authors.

(2) The ablation studies explain the role of norm and angular separations very well, and shows why the R2G loss efficiently combines information about both radial and angular separations.

(3) The results hold for a non-modular arithmetic task, showing that the R2G loss is relevant in settings beyond periodic maps as in modular arithmetic .

Weaknesses:

(1) The paper's key argument is that grokking occurs due to competition between norm increase and angular separation. Such arguments were already made in [6], but was subsequently contested in [2] and [3]. It would be helpful for the authors to devote discussions how their paper differs from these past works.

(2) I was a little confused why the authors chose the reciprocal of the the term $\sum_{i=1}^{n} ||v_i - v_i'||^2$. It seems to be more unstable especially if the vectors themselves become very small during training, when it can easily overwhelm the actual cross-entropy loss. Some regularization in the denominator seems necessary. Alternately, the authors should do a proper stability analysis (and not just a robustness analysis of $\alpha$).

(3) The observation that embeddings form structured representations has been reported before [1], [5]. The authors need to acknowledge these results.

(4) Missing reference to grokking representations studied in [4]

(5) Possible typo in Fig. 1 (d) : Validation/test instead of Train/test? It should be clarified what exactly the "ratio" means in subplots (c) and (f).

(6) Although the authors acknowledge that previous works have focussed on understanding the learnt features, the proposed R2G loss itself is motivated by post-training representations. In this regard, the authors do not actually present a first-principles motivation for R2G loss, in conflict with their claims in the introduction. Thus, this loss maybe only relevant for settings where grokking is observed.

[1] Liu, Z., Kitouni, O., Nolte, N., Michaud, E. J., Tegmark, M., & Williams, M. (2022). Towards understanding grokking: An effective theory of representation learning. arXiv. https://arxiv.org/abs/2205.10343

[2] Golechha, S. (2024). Progress measures for grokking on real-world tasks. arXiv. https://arxiv.org/abs/2405.12755

[3] Minegishi, G., Iwasawa, Y., & Matsuo, Y. (2025). Bridging lottery ticket and grokking: Understanding grokking from inner structure of networks. arXiv. https://arxiv.org/abs/2310.19470

[4] Gromov, A. (2023). Grokking modular arithmetic. arXiv. https://arxiv.org/abs/2301.02679

[5] He, T., Doshi, D., Das, A., & Gromov, A. (2024). Learning to grok: Emergence of in-context learning and skill composition in modular arithmetic tasks. arXiv. https://arxiv.org/abs/2406.02550

[6] Liu, Z., Michaud, E. J., & Tegmark, M. (2023). Omnigrok: Grokking beyond algorithmic data. arXiv. https://arxiv.org/abs/2210.01117

---

> ### Author Rebuttal · Authors · 2026-03-30
>
> Thank you for the thoughtful and detailed feedback, which has helped us significantly improve the quality of our paper.
>
> ---
> **Response to Weakness (1):**
>
> Our analysis operates at a different level from [2], [3], and [6]: rather than examining model weight norms, we analyze the angular geometry of token embeddings in the Value space of the attention mechanism. Beyond this analysis, R2G loss provides a direct, active intervention on embedding geometry, causally inducing the angular reorganization that drives grokking. Crucially, R2G induces *phase transitions* rather than merely reducing delay, a capability absent from [2], [3], and [6]. We will add an explicit discussion of these distinctions in the revised manuscript.
>
> ---
> **Response to Weakness (2) and Key Question (2):**
>
> We fully agree that a rigorous analysis of the random permutation mechanism is essential for a complete theoretical account of R2G. We have substantially expanded the theoretical analysis to address these questions:
>
> Let $\{v_1, \ldots, v_B\}$ be a batch of Value vectors, and let $\pi \sim \text{Uniform}(\mathcal{S}_B)$ be a uniformly random permutation. Define the single-sample distance sum:
>
> $$D_\pi = \sum_{i=1}^B||v_i - v_{\pi(i)}||_2$$
>
> Since the marginal distribution of $\pi(i)$ is uniform over $\{1, \ldots, B\}$, we have:
> $$
> E_\pi[D_\pi] = \sum_{i=1}^B \frac{1}{B} \sum_{j=1}^B||v_i - v_j|| = \frac{2}{B} \sum_{i < j} ||v_i - v_j||
> $$
> Therefore:
> $$
> E_\pi[D_\pi] = (2/B) * D_{sum}, \quad \text{where} \quad D_{sum} = \sum_{i<j} ||v_i - v_j||
> $$
>
> This shows that **in expectation, maximizing $D_\pi$ is equivalent to maximizing the total pairwise distance sum** $\mathcal{D}_\text{sum}$, whose optimal solution is a uniform distribution of points on the hypersphere, matching the circular embedding topology observed in the grokking phase.
>
> Since $f(x)=1/x$ is convex, Jensen's inequality gives
> $$
> E_\pi[ \frac{\lambda}{D_\pi} ] \geq \frac{\lambda}{E_\pi[ D_\pi ]}
> $$
> with equality only when $D_\pi$ is constant across permutations. A second-order Taylor expansion shows that the gap is approximately $\sigma_{D}^2/\bar{D}^3$. Crucially, $Var(D_\pi)=\sigma_D^2$ is *large* when the embedding batch contains collapsed vectors (memorization) and *small* when embeddings are already well-dispersed (grokking). Consequently, R2G automatically applies a stronger repulsive force to collapsed configurations.
>
> ---
> **Response to Key Question (1):**
>
> The reciprocal form is necessary, and we explain this from two angles.
>
> *Theoretically*, the key difference between $\mathcal{L}=1/D$ and alternatives such as $-D$ or $-D^2$ lies in **adaptive gradient scaling**. When embeddings collapse ($D$ small, memorization), the $1/D^2$ factor in $\nabla_{v_k} \mathcal{L}$ automatically amplifies the repulsive force; when embeddings are already well-separated ($D$ large, near grokking), the force diminishes, avoiding disruption of the already-formed circular topology. By contrast, the $-\sum||v_i - v'_i||^2$ form produces gradients proportional to $-(v_k - v'_k)$, whose magnitude *grows* with distance, applying stronger repulsion when vectors are already well-separated, which risks destroying the structured geometry in the grokking phase.
>
> *Empirically*, we validated this by comparing two forms: R2G-sq ($-\lambda \sum ||v_i - v_i'||^2$) and R2G-linear ($-\lambda \sum ||v_i - v_i'||$). Both alternatives fail to grokking completely: the auxiliary loss term grows uncontrollably during training. In contrast, monitoring the R2G loss value shows it starts at approximately $4.26$ and decays toward $0$ as training progresses, is precisely what we proof.
>
> Regarding the reviewer's concern about numerical instability, we first verified empirically that $D_\text{train}$ remains reasonable throughout training. We also ran experiments adding a small constant $\epsilon$ to the denominator and found no meaningful difference in outcomes. Our conclusions remain unchanged with or without $\epsilon$, though we agree this is a sensible safeguard and thank the reviewer for raising it.
>
> ---
> **Response to Weakness (6):**
>
> We acknowledge that R2G's design is informed by observations of post-training representations. However, the theoretical analysis in Section 6 provides an *independent* motivation that does not rely on observing grokked representations. In fact, the effectiveness of our method has sufficiently strong causal support. In response to Reviewer 5oZr's comments, we have conducted a thorough causal analysis and experimental validation.
>
> ---
> **Response to Weakness (3)(4)(5):**
>
> Finally, on the question of generality, grokking is validated on different tasks and model architectures. We have elaborated on its generality and potential in our responses to Reviewers HxXb and 5oZr. We will also address all of the problem in (3), (4), (5) in the revised manuscript.
>
> ---
> We hope that the above explanations will allow us to raise the quality of the manuscript and receive a more favorable assessment.

---

> > ### Author Rebuttal · Reviewer_9CR9 · 2026-04-02
> >
> > I thank the authors for their detailed responses. I have a few follow-ups.
> >
> > (1) I thank the authors for explaining this discrepancy.
> >
> > (2) I thank the authors for the added theoretical discussion. However, I was mainly interested in knowing if there is any bias during training due to the random permutations of vectors in each batch. It might be difficult to perform experiments in this regard during the short time, so this will not substantially affect my judgement.
> >
> > (3) I'm quite concerned by the ablation experiment.  Does any $\lambda > 0$ cause such explosions? In the limit $\lambda \to 0$, you should be able to recover grokking without R2G. Then, the hyperparameter $\lambda$ may not be chosen correctly. Also it might be worthwhile to use some sort of layer normalization to control such uncontrolled norm growth. Another aspect might be to utilize constant weight norms, as introduced in https://whenwen.github.io/wd_blog/public/hyperball-part-1.html.
> >
> > The reason I'm emphasizing a lot on these points is because of the well-known issue of outlier sensitivity for such reciprocal losses.
> >
> > (4) I thank the author for the causal experiments. It is still worthwhile to emphasize this point in limitations.

---

> > > ### Author Response · Authors · 2026-04-04
> > >
> > > We sincerely thank you for the continued engagement. We address each point carefully below.
> > >
> > > ---
> > > **On the stability concern**
> > >
> > > We appreciate the focus on this issue, it has led us to a more complete analysis. We provide a theoretical analysis showing that the repulsion term in $\mathcal{L}_{R2G}​$ is provably stable.
> > >
> > > For a single pair $(v_i, v_j)$ with distance $r = ||v_i - v_j||=||\delta||$, the gradient of the repulsion term drives:
> > >
> > > $$\dot{r} = -\hat{\delta}^\top g_{ij} + \frac{4\lambda}{r^3}$$
> > >
> > > where $g_{ij} = \nabla_{v_i}L_\text{task} - \nabla_{v_j}L_\text{task}$ is the relative task gradient, and $\hat{\delta} = \delta/r$ is the unit direction. Since $|{\hat{\delta}^\top g_{ij}}| \leq B$ (bounded task gradient), whenever $r < (4\lambda/B)^{1/3}$, we have $\dot{r} > 0$, so the system is always pushed outward. The singularity at $r=0$ is not an attractor but a repulsive wall.
> > >
> > > High-dimensional geometry further reduces risk. Under standard initialization $v \sim \mathcal{N}(0, \sigma^2 I_d)$, the pairwise distance satisfies $||v_i - v_j||^2 \sim 2\sigma^2\chi^2_d$. The probability of two vectors being unusually close decays as:
> > >
> > > $$P(||v_i - v_j|| \leq r) \approx \frac{1}{\Gamma(d/2+1)}\left(\frac{r^2}{4\sigma^2}\right)^{d/2}$$
> > >
> > > which is exponentially small in $d$. With $d=48$ in our setting, near-collision is essentially impossible in practice.
> > >
> > > ---
> > > **Response to Question (3)**
> > >
> > > As established in our previous response, the reciprocal form provides adaptive gradient scaling. The two alternatives behave oppositely: For the loss $-\lambda D^2$, the gradient is proportional to $|v_k - v'_k|$, which grows as embeddings separate. This is a structural flaw, not a hyperparameter issue. We verified this by sweeping $\lambda \in {10^{-6}, 10^{-4}}$: at small $\lambda$ the training loss still diverges.
> > >
> > > Following your suggestion, we tested the normalization methods, LayerNorm and Hyperball. **LayerNorm** controll the absolute scale while preserving relative structure. After LayerNorm, the quadratic form loss gradient scales as $O(\sin\theta_{kk'})$, which still vanishes at collapse ($\theta \to 0$), providing less repulsion when it is needed. Empirically, it achieves a grokking ratio of **60%**, compared to **73%** for R2G (**https://anonymous.4open.science/r/anonymous-figure-icml/figures.pdf Fig.5(a)**).
> > >
> > > **Hyperball** projects all vectors onto the unit hypersphere. We note that this is equivalent to our Angular-only ablation already reported in the paper. Empirically, quadratic loss under Hyperball only achieves a grokking ratio of **33%**, still demonstrate the limitations of the Angular-only loss (**Fig 5(b)**).
> > >
> > > These results confirm that the reciprocal form has more advantages.
> > >
> > > ---
> > > **Response to Question (2)**
> > >
> > > Regarding your concern that random permutation affects the optimization, we now provide a complete characterization.
> > >
> > > First, we note that in our training procedure, the V-vector pairs are re-sampled at every epoch across 18,000 epochs, ensuring sufficient random coverage in practice.
> > >
> > > In each training step, we sample a random permutation $\pi \sim \text{Uniform}(\mathcal{S}_B)$ and compute pairwise distances only between matched pairs $(v_i, v\_{\pi(i)})$. Although this involves only $B$ pairs rather than all $\binom{B}{2}$ pairs, it is an efficient and unbiased sampling strategy. Formally, by the linearity of expectation:
> > >
> > > $$\mathbb{E}\_{\pi}[D\_{\pi}] = \frac{2}{B}\sum_{i<j}||v_i - v_j|| = \frac{2}{B}\mathcal{D}_\text{sum}$$
> > >
> > > and since
> > > $$\nabla\_V \mathbb{E}\_{\pi}[\mathcal{L}\_{R2G}] = \mathbb{E}\_{\pi}[\nabla\_V \mathcal{L}\_{R2G}]$$
> > >
> > > under standard regularity conditions, the expected gradient is an unbiased estimator of the gradient of the full pairwise objective. The optimization direction is therefore guaranteed to be correct on average, the random permutation does not bias the model away. This is ensured by the law of large numbers. So no systematic directional bias is introduced.
> > >
> > > While there is no bias, your concern about randomness is well-placed: although the direction is correct on average, each individual step introduces variance. We now show this variance is well-controlled.
> > >
> > > Assume pairwise distances are bounded by $\Delta$. One can show $\text{Var}(D\_\pi) \leq \frac{B\Delta^2}{2}$, so the coefficient of variation satisfies:
> > > $$\frac{\text{Std}(D\_\pi)}{\mathbb{E}[D\_\pi]} = O\!\left(B^{-1/2}\right)$$
> > > Once embeddings begin to disperse, the variance of the loss itself further satisfies $\text{Var}(\mathcal{L}\_{R2G}) = O\!(B^{-3})$, decaying cubically in batch size. Since we use full-batch training, the permutation noise is effectively negligible, guaranteeing algorithmic stability.
> > >
> > > ---
> > > **Response to Question (4)**
> > >
> > > Following your suggestion, we will discuss the limitations of causal interpretation more explicitly in the limitations section.
> > >
> > > ---
> > > We will incorporate the above analyses into the paper and appendix, and hope to have the opportunity to revise the paper accordingly.

---

### Official Review · Reviewer_5oZr · 2026-03-12

**Soundness:** 3
**Presentation:** 3
**Significance:** 3
**Originality:** 3
**Overall Recommendation:** 4
**Confidence:** 2

**Summary:**

This paper investigates grokking, the phenomenon where models suddenly generalize long after overfitting training data. It shows that grokking arises from the structural and geometric reorganization of token embeddings. Based on this, the authors propose the R2G loss to reshape the embedding manifold. Experiments on algorithmic and linguistic tasks validate its effectiveness.

**Compliance With Llm Reviewing Policy:**

Affirmed.

**Final Justification:**

The author addressed my core concern. I recommend accepting this paper. But I am not quite familiar with this field which is reflected in my confidence score. I am not sure if this paper is a clear accept. If AC leans toward rejection, I will defer to AC's decision.

**Key Questions For Authors:**

1. As noted in Weakness 1, could the authors provide additional causal evidence to support their mechanistic claims?

2. Including at least one experiment about modern network, such as deeper Transformers, more attention layers, or more complex language tasks. This would improve the paper.

**Limitations:**

Yes

**Strengths And Weaknesses:**

Strength:
1. The paper focus on an important problem. Grokking remains a key phenomenon for understanding the generalization dynamics of DNNs, and connecting the sudden late-phase generalization during training to the geometry of internal representations is potentially valuable.

2. The paper is well written, with a clear overall structure and logical organization.

Weakness：
1. Strictly speaking, the mechanistic evidence provided in the paper is mainly based on correlation, rather than sufficiently strong causal proof. The claim of “The Geometric Origin” in the title is relatively stronger, and it seems that the current evidence is not adequate to fully support such a strong causal statement. Additional causal evidence would help strengthen the paper.
2. According to Section 3.3, the experimental scope is relatively limited. The model is a very shallow, single-layer decoder-only Transformer (4 heads, d_model=48). In Section 5.2, the arithmetic tasks focus mainly on modular addition and multiplication, and only one tense-inflection benchmark is used for language tasks.

---

> ### Author Rebuttal · Authors · 2026-03-30
>
> We sincerely thank the reviewer for the thoughtful and encouraging feedback. We are glad the reviewer finds the paper well-written and the problem important. We address each concern below.
>
> ---
> ### **1.  Causal Evidence for the Geometric Origin of Grokking**
>
> We appreciate the reviewer's precise identification of the distinction between correlation and causation. We agree that the title's causal claim warrants stronger support, and we would like to clarify that our evidence spans all three rungs of Pearl's *Ladder of Causation* (*The Book of Why*, 2018):
>
> **Step 1 — Association** ($P(Y|X)$): Our mechanistic analysis (Figures 2–4 in our paper) documents a consistent correlation between V-vector geometric reorganization and the onset of grokking across training phases, establishing the observational foundation.
>
> **Step 2 — Intervention** ($P(Y|\text{do}(X))$): R2G Loss itself constitutes a controlled do-operator intervention. Under strictly identical data, model, and optimizer conditions, we directly manipulate the embedding geometry and observe a corresponding change in generalization behavior. If geometric reorganization were merely a by-product of grokking rather than its cause, actively enforcing it should not induce grokking — yet it consistently does, falsifying the null hypothesis.
>
> **Step 3 — Counterfactual** ($P(Y_x | X', Y')$): To further probe the causal direction, we conducted a **geometric disruption experiment**: starting from models already in the grokking phase, we injected additive random perturbations into token embeddings at different epochs (400, 600, 800, 1000) to deliberately destroy the geometric structure, and monitored validation accuracy under varying perturbation scales.
>
> | noise\_scale | timing | recover\_rate |
> |:---:|:---:|:---:|
> | 0.1 | before\_rise | 60% |
> | 0.1 | after\_rise | 100% |
> | 1.0 | before\_rise | **0%** |
> | 1.0 | after\_rise | **80%** |
> | 2.0 | before\_rise | 0% |
> | 2.0 | after\_rise | 0% |
>
> ("Timing" in the sheet indicates whether the perturbation occurs before grokking or after grokking.)
>
> The results reveal a clear causal signature: the timing of disruption relative to the validation accuracy rise proves decisive: geometric structure becomes progressively more resistant to perturbation as training advances, and models closer to full grokking are better able to withstand and self-repair from identical perturbation strengths. This monotonic consolidation of geometric structure, which stabilizes before the sharp rise in validation accuracy, establishes a clear temporal precedence: the formation of geometric structure precedes, and causally drives, the emergence of generalization.
>
> Together, these three levels of evidence: observation, intervention, and counterfactual disruption, provide a principled causal account consistent with the do-calculus framework.
>
> The above discussion on causality and the supplementary experiments will be included in the appendix of the revised manuscript.
>
> ---
>
> ### **2. Experimental Scope and Model Diversity**
>
> We thank the reviewer for this suggestion, and would like to clarify a point that may not have been immediately apparent from the main text.
>
> As the reviewer correctly notes, Section 3.3 describes a 1-layer decoder-only Transformer ($d_{\text{model}}=48$, 4 heads) used for the modular arithmetic analysis. However, as detailed in **Appendix B.1**, the Tense-Inflection experiment in Section 5.3 uses a substantially different architecture: a **4-layer encoder-only Transformer** with 8 attention heads and $d_{\text{model}}=512$. We acknowledge this was not made sufficiently clear in the main text and will add an explicit note in Section 3.3 in the revision.
>
> To directly address the concern about architectural generality, we also conducted an additional experiment on a **2-layer decoder-only Transformer** on the $P=113$ modular addition task. R2G remains highly effective, lifting the grokking rate from **7% to 73%** (see **https://anonymous.4open.science/r/anonymous-figure-icml/figures.pdf - Fig 4**), confirming that the acceleration effect persists as model depth increases.
>
> Also, we have elaborated on R2G’s effectiveness, generality and potential in responses to reviewers HxXb. We have additionally conducted comparative experiments with methods (Grokfast and NeuralGrok), demonstrating that our R2G loss offers unique value in promoting the phase transition process of grokking. We also performed a sweep over the weight decay parameter, showing that the effectiveness of R2G is robust and not affected by the choice of weight decay. And we further observed the phenomenon of V-vector structural reorganization even in LLM (OLMo), along with a preliminary analysis of this phenomenon. All result figures are provided in https://anonymous.4open.science/r/anonymous-figure-icml/figures.pdf.
>
> ---
>
> We thank you again for the constructive and fair assessment, and look forward to incorporating these improvements into the revised manuscript.

---

> > ### Author Rebuttal · Reviewer_5oZr · 2026-04-02
> >
> > Thank the author for the detailed rebuttal. I would like to maintain the positive score.

---

### Official Review · Reviewer_HxXb · 2026-03-13

**Soundness:** 2
**Presentation:** 3
**Significance:** 2
**Originality:** 2
**Overall Recommendation:** 4
**Confidence:** 5

**Summary:**

This paper investigates the geometric mechanisms underlying grokking in neural networks. The authors analyze how token embeddings in Transformer attention layers reorganize across training phases (memorization -> semi-grokking -> grokking), observing that value ($V$) vectors — but not query ($Q$) or key ($K$) vectors — transition from collapsed clusters to structured circular topologies with maximal separation between numerically adjacent tokens. Based on this observation, they propose R2G (Repel-to-Grokking) Loss, which maximizes pairwise Euclidean distances between randomly paired $V$ vectors to encourage this structural separation. They provide a theoretical decomposition of embedding gradients into radial and angular components, arguing that angular reorganization is the slow bottleneck primarily driving grokking.

**Compliance With Llm Reviewing Policy:**

Affirmed.

**Final Justification:**

The rebuttal addressed my three main concerns with substantial additional experiments. The Grokfast/NeuralGrok comparison convincingly demonstrates R2G's advantage in both grokking rate and convergence speed. The weight decay sweep confirms that R2G provides an independent geometric benefit across all settings, including wd=0. The OLMo-2-1B preliminary analysis and the 2-layer decoder experiment extend the paper's findings beyond the original shallow Transformer setting. These results collectively resolved my concerns and shifted my evaluation from 3 (weak reject) to 4 (weak accept).

**Key Questions For Authors:**

Please see weaknesses. I'm willing to revise my score once these are carefully addressed.

**Limitations:**

Yes

**Strengths And Weaknesses:**

**Strengths**
- The paper presents a well-structured empirical pipeline that consistently supports its central claim: grokking is driven by angular reorganization rather than norm growth.
- The writing is clear, and the visual narrative is effective. The progression from mechanistic analysis to intervention design to theoretical justification makes the overall argument easy to follow.
- The radial-angular decomposition leads to a testable and falsifiable prediction, which is validated through ablation studies. This framing could also inform future work on geometry-aware training objectives beyond the grokking setting.
- Although the geometric observation of circular embeddings in grokking builds on established prior findings, the paper’s main contribution is to turn that observation into an active, lightweight loss (R2G). By explicitly encouraging value-vector repulsion, this method offers a meaningful, though incremental, advance beyond earlier purely diagnostic work.

**Weaknesses**
- Recent work on accelerating grokking includes Grokfast, GrokTransfer, NeuralGrok, and lottery ticket pruning. Comparing R2G with one or two of these methods, especially Grokfast, would help readers better contextualize its benefits. Since R2G is geometrically motivated rather than based on gradient filtering or transfer, it may offer complementary advantages worth highlighting. It would also be valuable to compare the methods under a fixed training compute budget and ask which achieves the highest grokking success rate.
- Since both weight decay and R2G shape embedding geometry, it would be valuable to examine how they interact. Weight decay is known to be important for grokking (Power et al., 2022) and is fixed at wd = 0.01 throughout the experiments. Evaluating R2G across a range of weight decay values, including no weight decay, would clarify whether R2G provides an independent geometric benefit or mainly amplifies the regularization effect of weight decay. For a fair comparison, the results should also be evaluated using the best weight decay setting for each method. Running a weight decay sweep for each experiment, especially the Figure 1 setup, and reporting the best-performing weight decay for each random seed would make the comparison substantially more convincing.
- All mechanistic analyses use a 1-layer Transformer with d=48, and the extension to a 4-layer encoder for tense inflection is appreciated. A natural next step would be to test whether the V-vector reorganization dynamics also appear in larger models. Open-source LLMs with intermediate training checkpoints, such as OLMo, make it feasible to evaluate whether V-vector angular reorganization correlates with improvements in downstream reasoning benchmarks, without the prohibitive cost of training an LLM from scratch with R2G. Even a preliminary analysis would broaden the paper’s impact.

**Minor issues:**
- In References, Langley (2000) reference is the ICML LaTeX template example — it should be removed.
- Key quantitative claims use imprecise language ("improves for 80%", "decreases for 14%") — "by" would be correct.

---

> ### Author Rebuttal · Authors · 2026-03-30
>
> We sincerely thank the reviewer for the detailed and constructive feedback.
> Based on your suggestions, we have undertaken the following revisions to further improve the manuscript.
>
> ---
>
> ## 1. Comparison with Grokfast and NeuralGrok
>
> We have conducted additional experiments comparing **R2G** against **Grokfast** and **NeuralGrok** under identical training datasets on the modular addition task ($P = 500$, dataset size = 26,650).
>
> * **Grokfast** applies a low-pass filter (EMA) to the gradient stream to amplify slow, low-frequency components that drive generalization; we tuned its cutoff coefficient $\alpha$ to the task-optimal value of 0.3.
>
> * **NeuralGrok** trains an auxiliary neural amplifier that adaptively re-weights gradients; we held out a portion of training data for the amplifier and tuned its scale parameter $c_{\text{norm}} = 0.2$ and update frequency $inner  loop  steps = 2$ for optimal performance.
>
> Our experimental results show, under the same dataset:
>
> * R2G converts non-grokking runs into grokking with a gain of **73%**
> * Grokfast achieves **60%**
> * NeuralGrok achieves **27%**
>
> Moreover, among runs that successfully reach grokking, **R2G converges the fastest**.
>
> Visualization results (phase-distribution bar chart and epoch-wise grokking-rate curve) are provided in **https://anonymous.4open.science/r/anonymous-figure-icml/figures.pdf - Fig1**.
>
> These results highlight a qualitative distinction:
> Grokfast and NeuralGrok focus on **temporal acceleration of gradient signals**, whereas R2G directly **restructures the embedding manifold**, making it a complementary and more geometrically principled intervention for promoting phase transitions.
>
> We will include this comparison in the main text of revised manuscript.
>
> ---
>
> ## 2. Interaction of Weight Decay
>
> To determine whether R2G provides a geometric benefit independent of weight decay, we swept
> $\gamma \in {0,; 0.001,; 0.01,; 0.1}$ and evaluated both the baseline and R2G at each value.
>
> Our key findings are:
>
> 1. Even at $\gamma = 0$ (no weight decay), R2G still substantially promotes grokking (**Fig2(a)**), confirming that its effect is not mediated by weight decay.
>
> 2. R2G consistently improves over the baseline across *all* weight-decay settings (**Fig2**), with grokking-rate gains of:
>
>    * 66% ($\gamma = 0$)
>    * 67% ($\gamma = 0.001$)
>    * 73% ($\gamma = 0.01$)
>    * 20% ($\gamma = 0.1$)
>
> So even when each seed is evaluated at its optimal weight decay, R2G still outperforms the baseline.
>
> These results demonstrate that R2G provides an **independent geometric benefit** that is additive to, rather than reliant on, weight decay regularization.
>
> ---
>
> ## 3. Preliminary Analysis on Larger Pretrained Models (OLMo)
>
> To assess whether V-vector angular reorganization is a general phenomenon beyond shallow Transformers,
> we performed a **preliminary analysis** of **OLMo-2-1B** across publicly available pretraining checkpoints
> (9 checkpoints selected from step 0 to step 1M).
>
> We first extracted V-vectors for number tokens (0–99) across all 16 layers and computed the Pearson correlation $r$ between numerical difference $|i - j|$ and V-vector Euclidean distance (analogous to Figure 4 in the paper).
>
> The correlation increases monotonically in layer 0:
>
> > $r = -0.006$ (step 0) $\to$ $r = 0.453$ (step 50K) $\to$ $r = 0.750$ (step 1M)
>
> as shown in **Fig3(a)**.
>
> The remaining 15 layers exhibit the same trend, as shown in **Fig3(b)**.
>
> This confirms that numerical relationships are progressively encoded into the V-vector geometry during large-scale pretraining—consistent with our mechanistic findings on small Transformers.
>
> We further examined whether semantic relationships are encoded in V-vector geometry by grouping tokens into three categories (**animals, colors, numbers**) and computing the intra-/inter-group distance ratio $\rho = \bar{d}\_{\text{intra}} / \bar{d}\_{\text{inter}}$.
>
> We observe that $\rho$ decreases monotonically from approximately 1.0 at random initialization, indicating that semantically related tokens are progressively clustered into the same region of V-space, as shown in **Fig3(c)**.
>
> Together, these analyses suggest that V-vector structural reorganization is not an artifact of shallow architectures but a general phenomenon observable in large-scale pretraining, and provides promising potential for broader downstream applications.
>
> These results will be included in the appendix, accompanied by further discussion. We believe it provides valuable insights for extending our work in future research.
>
> ---
>
> Finally, we apologize for the oversight in leaving template-related references and for issues with wording accuracy.
> The spurious *Langley (2000)* reference will be removed. In addition, all instances of:
>
> * “improves/decreases **for** X%”
>
> will be corrected to:
>
> * “improves/decreases **by** X%”
>
> throughout the manuscript.
>
> ---
> We thank you again for the careful reading and look forward to the opportunity to revise the manuscript accordingly.

---

> > ### Author Rebuttal · Reviewer_HxXb · 2026-04-02
> >
> > I appreciate the authors' substantial effort in addressing my concerns with additional experiments and figures.
> >
> > * Comparison with Grokfast/NeuralGrok (Fig 1): The results convincingly demonstrate R2G's advantage. R2G achieves the highest grokking rate (73%) and fastest convergence, while notably NeuralGrok's failure rate (47%) is even worse than the baseline (13%). The qualitative distinction between geometric intervention (R2G) and gradient-based acceleration (Grokfast/NeuralGrok) is well articulated.
> > * Weight decay interaction (Fig 2): The sweep across wd=0, 0.001, and 0.1 is convincing. The fact that R2G provides a large improvement even at wd=0 confirms that its effect is not mediated by weight decay. At wd=0.1, the smaller gain is expected since the baseline already achieves strong grokking, which is consistent with the claim that R2G provides an independent, additive geometric benefit.
> > * OLMo analysis (Fig 3): The monotonic increase in Pearson ρ across all 16 layers of OLMo-2-1B during pretraining, together with the semantic clustering result, provides a meaningful preliminary analysis that extends the paper's findings beyond shallow Transformers. Connecting this to downstream task performance remains a promising direction for future work.
> > * 2-layer experiment (Fig 4): The additional result on a 2-layer decoder (baseline 7% → R2G 73%) further strengthens the depth generalization argument, going beyond what was originally requested.
> >
> > Given these results, I am revising my score upward from 3 to 4.

---

### Decision · Program_Chairs · 2026-04-30

**Decision:**

Accept (regular)

**Comment:**

The proposes a hypothesis that grokking occurs as the geometry of token embeddings change; the paper proposes a loss that intervenes and explicitly reshapes the embedding as a way of accelerating the grokking process and testing its hypothesis.

The reviews recognize the studied problem as important; indeed, grokking is of significant interest when it comes to understanding Transformer dynamics on algorithmic tasks. Multiple reviewers appreciate the narrative of the paper as well-structured (`HxXb`, `5oZr`) and well-written (`Ep2F`) and clear (`HxXb`). The proposed loss is interesting and original, and helps the paper go beyond just making observations about the geometry which past work has already done (`HxXb`, `9CR9`, `Ep2F`); the loss function is also interpretable (`9CR9`) and simple enough for others to build on (`Ep2F`).

Multiple concerns were resolved during the rebuttal, such as comparisons to baslines, which the authors seem to have addressed extensively. But the main concerns that remain are (a) the sensitivity/stability of the loss and (b) its generality to various settings and architecture. The author's response to (a), as per internal discussions, seems to have convinced nearly all reviewers positively; likewise, on architectural generality, the authors have added experiments that help this somewhat. We recommend that the authors include these discussions in the paper.

**There were a number of important pieces of related works the original submission has missed; we strongly urge the authors to do a more careful literature search and update the paper accordingly.**

Overall, we recommend accepting the paper since it provides a simple, novel and actionable intervention to understand a well-studied empirical phenomenon.